# A Layer-Wise Natural Gradient Optimizer for Training Deep Neural Networks

**Xiaolei Liu**[*]
Ant Group
Hangzhou, China
`liuxiaolei.lxl@mybank.cn`

**Shaoshuai Li**[*]
Ant Group
Hangzhou, China
`lishaoshuai.lss@mybank.cn`

**Kaixin Gao**[†]
Ocean University of China
Qingdao, China
`gaokaixin06@163.com`

**Binfeng Wang**
Ant Group
Hangzhou, China
`wangbinfeng.wbf@mybank.cn`

## Abstract

Second-order optimization algorithms, such as the Newton method and the natural gradient descent (NGD) method exhibit excellent convergence properties for training deep neural networks, but the high computational cost limits its practical application. In this paper, we focus on the NGD method and propose a novel layer-wise natural gradient descent (LNGD) method to further reduce computational costs and accelerate the training process. Specifically, based on the block diagonal approximation of the Fisher information matrix, we first propose the layer-wise sample method to compute each block matrix without performing a complete back-propagation. Then, each block matrix is approximated as a Kronecker product of two smaller matrices, one of which is a diagonal matrix, while keeping the traces equal before and after approximation. By these two steps, we provide a new approximation for the Fisher information matrix, which can effectively reduce the computational cost while preserving the main information of each block matrix. Moreover, we propose a new adaptive layer-wise learning rate to further accelerate training. Based on these new approaches, we propose the LNGD optimizer. The global convergence analysis of LNGD is established under some assumptions. Experiments on image classification and machine translation tasks show that our method is quite competitive compared to the state-of-the-art methods.

## 1 Introduction

With the rapid increase in the size of deep neural networks (DNNs) models in both areas of computer vision (CV) and natural language processing (NLP), there have been remarkable attentions given to optimizing algorithms. An effective optimizer can significantly improve the training speed of models while ensuring high prediction performance. First-order gradient descent methods are workhorses of training DNNs, which can be broadly divided into two categories: methods use a same learning rate, such as stochastic gradient descent (SGD) [1] and its accelerations [2, 3], and methods use adaptive learning rate, such as AdaDelta [4], RMSProp [5], ADAM [6] and Adabelief [7]. Although first-order gradient descent methods enjoy low computational cost and ease of implementation, they might suffer from sensitivity to hyperparameters and slow convergence. It is challenging to reduce the number of iterations and computational time of these methods.

---

[*]Joint first author, these authors contributed equally to this work.
[†]Corresponding author.

38th Conference on Neural Information Processing Systems (NeurIPS 2024).

Some work has considered introducing curvature information when updating parameters of DNNs to improve the convergence speed and overcome the above shortcomings of the first-order methods. However, second-order optimization methods need to store and compute the inverse of curvature matrix, which brings expensive storage and computation costs and limits the application of second-order methods in training large-scale DNNs. Therefore, many approximate second-order methods have been proposed for training large-scale models. For example, Keskar and Berahas [8] proposed a stochastic quasi-Newton algorithm for training recurrent neural networks. Yao et al. [9] approximated the Hessian matrix as a diagonal operator, which is achieved by applying Hutchinson's method, and proposed the AdaHessian method. Goldfarb, Ren and Bahamou [10] developed Kronecker-factored block-diagonal BFGS and its limited-memory variants L-BFGS methods for training DNNs. Generalized Gauss-Newton methods, such as the Hessian-free method [11] and the Krylov subspace method [12], also have been proposed to approximate the Hessian matrix.

The natural gradient descent (NGD) method [13], which preconditions the gradient by the Fisher information matrix instead of the Hessian matrix, also has shown effectiveness in training DNNs [14, 15, 16, 17]. NGD explores the steepest direction of the objective function when the parameter space has a Riemannian metric structure and has a faster convergence speed. In particular, NGD can also be seen as an approximation of the Netwon method when the objective function and the manifold metric are compatible [18]. However, it is still impossible to directly compute the inverse of the Fisher information matrix for DNNs with millions or even billions parameters. Quite a few approximate approaches have been proposed. Under some independency assumptions, Martens and Grosse [14] proposed the Kronecker-factored approximate curvature (KFAC) method, in which the Fisher information matrix is approximated as a block diagonal matrix and each block matrix is further approximated as the Kronecker product of two smaller matrices. Then, KFAC was extended to convolutional neural networks [19], recurrent neural networks [20] and variational Bayesian neural networks [21] and showed significant speedup during training. In addition, George et al. [22] proposed the eigenvalue-corrected Kronecker factorization (EKFAC) method. Gao et al. [15, 23] proposed the trace-restricted Kronecker-factored approximate (TKFAC) method. These approaches all focus on the Kronecker-factored approximations of the Fisher information matrix. What's more, some works have also considered large-scale distributed computing using NGD for training DNNs and shows excellent experimental performance [16, 24, 25].

In this paper, our main focus is on the NGD method. Motivated by the effectiveness of diagonal approximations and the significance of diagonal elements in the curvature matrix, we prioritize the diagonal information and integrate it into our approximation and introduce a novel method, namely Layer-wise Natural Gradient Descent (LNGD). Our contributions can be given as follows:

- Based on the block diagonal approximation of the Fisher information matrix, we propose a layer-wise sample method to more efficiently compute each block matrix corresponding to each layer. By assuming that the predictive distribution of the output after the activation function for each layer follows a Gaussian distribution, each block matrix can be directly computed using the inputs and the outputs separately, without having to perform a complete back-propagation.

- For each block matrix corresponding to each layer, we further approximate it as a Kronecker product of two smaller matrices, one of which is a diagonal matrix, while keeping the traces equal before and after approximation. With this operation, we further reduce the cost of computing inverse matrices while still preserving the main information of each block matrix.

- In order to further accelerate the training, we propose an adaptive layer-wise learning rate by optimizing a quadratic model, in which parameters in the same layer share the same adaptive learning rate. Moreover, a faster approach of computing the adaptive layer-wise learning rate is also provided, making it speed up training while maintaining computationally efficient.

- Based on the novel approximation mentioned above of the Fisher information matrix and the adaptive layer-wise learning rate, we propose the LNGD optimizer for training DNNs. The global convergence analysis are also established under some assumptions.

- We perform experiments on image classification and machine translation tasks. Numerical results show that LNGD converges faster than SGD, ADAM and KFAC, and LNGD provides an significant improvement in computational time savings when achieves convergence.

The rest of this paper is organized as follows. Section 2 gives the notations and introduces the NGD method. In Section 3, we propose a novel approximation of the Fisher information matrix and the

adaptive layer-wise learning rate. Furthermore, we give the framework of LNGD and establish the convergence analysis. Section 4 presents the results of experiments on image classification and machine translation tasks. The conclusion is drawn in Section 5.

## 2 Notations and Preliminaries

In this paper, for a matrix $\mathbf{A}$, we use $\mathbf{A}_{ij}$ to denote its $(i,j)$th entry, $\text{tr}(\mathbf{A})$ to denote its trace and $\|\mathbf{A}\|_{\mathbf{F}}$ to denote its Frobenius norm. We use $\circ$ and $\otimes$ to denote the Hadamard and Kronecker product of two matrices. In the following, we briefly introduce the NGD method for training DNNs. During the training process of neural networks, the purpose is to find the vector of parameters $\boldsymbol{\theta}$ which minimizes the loss function $h(\boldsymbol{\theta})$. If the loss function $h(\boldsymbol{\theta})$ is chosen as the the cross-entropy loss function, $h(\boldsymbol{\theta})$ can be given as $h(\boldsymbol{\theta}) = \mathbb{E}[-\log p(\mathbf{y}|\mathbf{x},\boldsymbol{\theta})]$, where $p(\mathbf{y}|\mathbf{x},\boldsymbol{\theta})$ is the density function of a predictive distribution $P_{\mathbf{y}|\mathbf{x}}(\boldsymbol{\theta})$, and $\mathbf{x}, \mathbf{y}$ are the training inputs and labels, respectively. Next, we give the definition of natural gradient, which gives the steepest direction of the objective function when the parameter space has a Riemannian metric structure. The natural gradient is defined as $\mathbf{F}^{-1}\nabla_{\boldsymbol{\theta}}\mathbf{h}(\boldsymbol{\theta})$, where $\mathbf{F}$ is the Fisher information matrix given by

$$\mathbf{F} = \mathop{\mathbb{E}}_{\mathbf{x}\sim q(\mathbf{x}), \mathbf{y}\sim p(\mathbf{y}|\mathbf{x},\boldsymbol{\theta})} [\nabla_{\boldsymbol{\theta}}\log p(\mathbf{y}|\mathbf{x},\boldsymbol{\theta})\nabla_{\boldsymbol{\theta}}\log p(\mathbf{y}|\mathbf{x},\boldsymbol{\theta})^{\top}]. \tag{1}$$

In Eq. (1), the input $\mathbf{x}$ is independently sampled from a distribution $Q_{\mathbf{x}}$ with density function being $q(\mathbf{x})$ and the label $\mathbf{y}$ is sampled from the predictive distribution $P_{\mathbf{y}|\mathbf{x}}(\boldsymbol{\theta})$. In the following pages, we abbreviate $\mathbb{E}_{\mathbf{x}\sim q(\mathbf{x}), \mathbf{y}\sim p(\mathbf{y}|\mathbf{x},\boldsymbol{\theta})}$ as $\mathbb{E}$ unless otherwise specified. Consider a neural network with $L$ layers, for each layer $l \in [L]$ with $[L] = \{1, 2, \ldots, L\}$, we denote $\mathbf{a}_{l-1}$ and $\mathbf{W}_l$ as the input (the activation from the previous layer) and the matrix of weights of this layer, respectively. What's more, $\boldsymbol{\theta}_l = \text{vec}(\mathbf{W}_l)$ and $\boldsymbol{\theta} = (\boldsymbol{\theta}_1, \ldots, \boldsymbol{\theta}_L)^{\top} = (\text{vec}(\mathbf{W}_1)^{\top}, \ldots, \text{vec}(\mathbf{W}_L)^{\top})^{\top}$, where $\text{vec}(\cdot)$ indicates vectorization of a matrix. For convenience, we denote the derivative of the loss function with respect to $\boldsymbol{\theta}$ as $\mathcal{D}\boldsymbol{\theta} = -\nabla_{\boldsymbol{\theta}}\log p(\mathbf{y}|\mathbf{x},\boldsymbol{\theta})$. Then the Fisher information matrix can be expressed as $\mathbf{F} = \mathbb{E}[\mathcal{D}\boldsymbol{\theta}\mathcal{D}\boldsymbol{\theta}^{\top}]$.

Due to the high computational and storage costs caused by the inverse operation of high-dimensional matrices, it is impractical to directly compute $\mathbf{F}^{-1}$ in the training of DNNs. The family of Kronecker-factored approximations provides an effective approach for computing $\mathbf{F}^{-1}$ of parameters in high-dimensional space, which is usually achieved by two steps. In the first step, by assuming that the parameters between different layers are independent, these methods approximate the entire Fisher information matrix as a block diagonal matrix, i.e.,

$$\mathbf{F} \approx \text{diag}(\mathbf{F}_1, \mathbf{F}_2, \ldots, \mathbf{F}_L), \tag{2}$$

where $\mathbf{F}_l = \mathbb{E}[\mathcal{D}\boldsymbol{\theta}_l\mathcal{D}\boldsymbol{\theta}_l^{\top}]$ for any $l \in [L]$. By this way, the Fisher information matrix can be approximated by $L$ block matrices. This step transforms the inverse of the entire Fisher information matrix into the inverse of a series of small block matrices. In the second step, these methods further approximate each block matrix as the Kronecker product of some smaller factors. This approximation can transform the inverse of each block matrix into the inverse of some smaller factors combining the properties of the Kronecker product.

## 3 LNGD: A Layer-Wise Second-Order Optimizer

In this section, we first introduce the layer-wise sample approximation strategy. Then, we present the details of adaptive layer-wise learning rate mechanism and give the specific framework of LNGD. Finally, elaborate theoretical analysis of LNGD's convergence is also provided.

### 3.1 Layer-Wise Sample Approximation

For NGD methods to train DNNs, the Fisher information matrix can be approximated by a block diagonal one according to different layers as given by Eq. (2), this approximation can be found in [14, 15, 19, 22] and references therein. We call such a block diagonal approximate Fisher information matrix the layer Fisher information matrix, which is computed based on a distribution $Q_{\mathbf{x}}$ and a predictive distribution $P_{\mathbf{y}|\mathbf{x}}(\boldsymbol{\theta})$ as given in Eq. (1). To obtain the layer Fisher information matrix, we

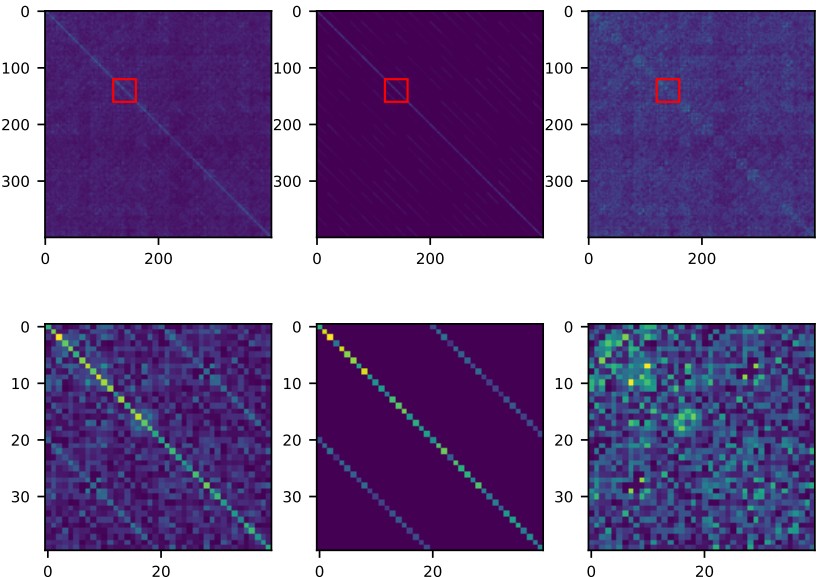

Figure 1: Comparison of the exact Fisher information matrix $\mathbf{F}$ and our approximation $\mathbf{F}_{\text{LNGD}}$. We use LNGD to train MNIST on a fully-connected neural network, whose architecture is 196-20-20-20-20-10. We show the results of the Fisher information matrix of the first layer with 20 units in top, which is a $400 \times 400$ matrix. The bottom portion displays partially enlarged parts of the top marked with red square, which is a $40 \times 40$ matrix. Within both the top and bottom sections, on the left is the exact Fisher information matrix $\mathbf{F}$, in the middle is our approximation $\mathbf{F}_{\text{LNGD}}$, and on the right is the absolute error between them. The brightness levels correspond to the sizes of the absolute values.

need perform a complete back-propagation to sequentially compute $\mathbf{F}_L, \mathbf{F}_{L-1}, \ldots, \mathbf{F}_1$, which still consumes much computing time.

In this subsection, we propose a layer-wise sample approximation of the Fisher information matrix, in which each block matrix $\mathbf{F}_l$ is computed based on the $l$th layer's prediction distribution $P_{\mathbf{a}_l|\mathbf{a}_{l-1}}(\boldsymbol{\theta}_l)$ with the input $\mathbf{a}_{l-1} \in \mathbb{R}^{d_l}$ of this layer and the input $\mathbf{a}_l \in \mathbb{R}^{d_{l+1}}$ of the $(l+1)$th layer instead of using the same predictive distribution $P_{\mathbf{y}|\mathbf{x}}(\boldsymbol{\theta})$ for all layers. Specifically, for $\mathbf{a}_l$, we assume that the predictive distribution $P_{\mathbf{a}_l|\mathbf{a}_{l-1}}(\boldsymbol{\theta}_l)$ follows Gaussian distribution, which is usual used as prior by variational auto-encoder [26], so $\mathbf{F}_l$ can be computed by sampling from a normal distribution with expectation being $\mathbf{a}_l$ and variance being $\mathbf{I}$. Similar assumption can also be found in [27], in which the normality is also supported by a central limit theorem under the independence assumption. By this layer-wise sample approximation, we can compute the layer Fisher information matrix without having to perform a complete back-propagation and thus improve the computational efficiency.

Next, we can give the formula of each block $\mathbf{F}_l$ in the layer Fisher information matrix as

$$\mathbf{F}_l = \mathbb{E}[\tilde{\mathcal{D}}\boldsymbol{\theta}_l\tilde{\mathcal{D}}\boldsymbol{\theta}_l^\top] = \mathbb{E}[\text{vec}(\tilde{\mathcal{D}}\mathbf{W}_l)\text{vec}(\tilde{\mathcal{D}}\mathbf{W}_l)^\top] = \mathbb{E}[\text{vec}(\mathbf{g}_l\mathbf{a}_{l-1}^\top)\text{vec}(\mathbf{g}_l\mathbf{a}_{l-1}^\top)^\top]$$
$$= \mathbb{E}[(\mathbf{a}_{l-1}\mathbf{a}_{l-1}^\top) \otimes (\mathbf{g}_l\mathbf{g}_l^\top)] \in \mathbb{R}^{m_l d_l \times m_l d_l},$$

where $\tilde{\mathcal{D}}\boldsymbol{\theta}_l = -\nabla_{\boldsymbol{\theta}} \log p(\mathbf{a}_l|\mathbf{a}_{l-1}, \boldsymbol{\theta}_l), \mathbf{g}_l = -\nabla_{\mathbf{s}_l} \log p(\mathbf{a}_l|\mathbf{a}_{l-1}, \boldsymbol{\theta}_l) \in \mathbb{R}^{m_l}$ with $\mathbf{s}_l = \mathbf{W}_l\mathbf{a}_{l-1}$, and $p(\mathbf{a}_l|\mathbf{a}_{l-1}, \boldsymbol{\theta}_l)$ is the density function of the distribution $P_{\mathbf{a}_l|\mathbf{a}_{l-1}}(\boldsymbol{\theta}_l)$.

In practice, the dimension of each block matrix $\mathbf{F}_l$ is often still too large to directly compute its inverse matrix. Therefore, additional approximation methods are required to handle this computational difficulty. Suppose that the predictive distribution of $\mathbf{a}_l$ follows Gaussian distribution with expectation being $\mathbf{a}_l$ and variance being $\mathbf{I}$, and each element of activation output $\mathbf{a}_l$ is independent and identically distributed random number, then each element of partial derivative $\mathbf{g}_l$ is also independent and identically distributed. It is easy to show that $\mathbf{F}_l$ can be seen as a matrix with $d_l \times d_l$ block matrices, in which each block is an $m_l \times m_l$ matrix and the off-diagonal elements are zero. Therefore, $\mathbf{F}_l$ can be approximated as

$$\mathbf{F}_l \approx \mathbb{E}[(\mathbf{a}_{l-1}\mathbf{a}_{l-1}^\top) \otimes \text{diag}(\mathbf{g}_l\mathbf{g}_l^\top)]. \tag{3}$$

Combining the property that $(\mathbf{A} \otimes \mathbf{B})^{-1} = \mathbf{A}^{-1} \otimes \mathbf{B}^{-1}$ for any two invertible matrices $\mathbf{A}$ and $\mathbf{B}$, we can significantly reduces the computational complexity. Thus, some approaches have considered approximating the Fisher information matrix as the Kronecker product of two factors [14, 15, 19, 20, 22]. Inspired by these works, we also approximate $\mathbf{F}_l$ as the Kronecker product of two factor matrices $\mathbf{\Phi}_l \in \mathbb{R}^{d_l \times d_l}$ and $\mathbf{\Psi}_l \in \mathbb{R}^{d_l \times d_l}$. To get factor matrices $\mathbf{\Phi}_l$ and $\mathbf{\Psi}_l$, we first replace $\mathrm{diag}(\mathbf{g}_l \mathbf{g}_l^\top)$ in Eq. (3) by its trace and obtain $\mathbf{\Phi}_l$. Then we compute $\mathbf{\Psi}_l$ while keeping that $\mathrm{tr}(\mathbf{F}_l) = \mathrm{tr}(\mathbf{\Phi}_l \otimes \mathbf{\Psi}_l)$. Specifically, $\mathbf{\Phi}_l$ is given by

$$\mathbf{\Phi}_l = \mathbb{E}[(\mathbf{a}_{l-1}\mathbf{a}_{l-1}^\top) \otimes \mathrm{tr}(\mathrm{diag}(\mathbf{g}_l\mathbf{g}_l^\top))] = \mathbb{E}[(\mathbf{a}_{l-1}\mathbf{a}_{l-1}^\top) \times \mathbf{g}_l^\top\mathbf{g}_l], \tag{4}$$

On the other hand, $\mathbf{\Psi}_l$ can be computed by

$$\mathbf{\Psi}_l = \frac{\mathbb{E}[(\mathbf{a}_{l-1}^\top\mathbf{a}_{l-1}) \times \mathrm{diag}(\mathbf{g}_l\mathbf{g}_l^\top)]}{\mathbb{E}[(\mathbf{a}_{l-1}^\top\mathbf{a}_{l-1})(\mathbf{g}_l^\top\mathbf{g}_l)]}. \tag{5}$$

Based on Eq. (4) and Eq. (5), we can show that $\mathrm{tr}(\mathbf{F}_l) = \mathrm{tr}(\mathbf{\Phi}_l \otimes \mathbf{\Psi}_l)$.

Fig. 1 presents the visualization results of the exact Fisher information matrix $\mathbf{F}$, our approximation $\mathbf{F}_{\mathrm{LNGD}}$, and the absolute error between them. Brighter pixels indicate higher values. From the left column in the top row, we observe the elements in the principal diagonal exhibit quite higher values, indicating their significance with rich information. Similarly, $\mathbf{F}_{\mathrm{LNGD}}$ can also emphasize the importance of the diagonal elements. The error figure reveals that the errors of the diagonal elements are small, which indicates that $\mathbf{F}_{\mathrm{LNGD}}$ provides a good approximation effect for the diagonal elements. Furthermore, to achieve a clearer visualization, we show the results of the partially enlarged area marked with red square in the bottom row. Here, we can observe more clearly that $\mathbf{F}_{\mathrm{LNGD}}$ achieves a favorable approximation effect on the diagonal elements. What's more, $\mathbf{F}_{\mathrm{LNGD}}$ can also provide an effective approximation of the elements in the auxiliary diagonals. These visualizations demonstrate the effectiveness of our proposed approximation in capturing the main elements of the Fisher information matrix. Therefore, our proposed approximation $\mathbf{F}_{\mathrm{LNGD}}$ is efficient and $\mathbf{F}_{\mathrm{LNGD}}$ can retain most of information.

## 3.2 Adaptive Layer-Wise Learning Rate

In this subsection, we propose an adaptive layer-wise learning rate to accelerate training DNNs. We first consider the cases that use the same learning rate for all elements and the adaptive element-wise learning rate. Then we present the adaptive layer-wise learning rate scheme.

Suppose that $\mathbf{d}^k$ is the update direction of the function $h : \mathbb{R}^n \to \mathbb{R}$ at the iteration point $\boldsymbol{\theta}^k$. We first recall the gradient descent methods for getting the minimization of $h$, in which the update rule can be given as $\boldsymbol{\theta}^{k+1} = \boldsymbol{\theta}^k - \alpha^k \mathbf{d}^k$, where $\alpha^k$ is the learning rate, which can be chosen according to the value of the quadratic model

$$h(\boldsymbol{\theta}^k - \alpha^k\mathbf{d}^k) \approx h(\boldsymbol{\theta}^k) - \alpha^k\langle\mathbf{d}^k, \nabla_{\boldsymbol{\theta}}h(\boldsymbol{\theta}^k)\rangle + \frac{(\alpha^k)^2}{2}(\mathbf{d}^k)^\top\nabla_{\boldsymbol{\theta}}^2 h(\boldsymbol{\theta}^k)\mathbf{d}^k.$$

Once the update direction is chosen, the minimizer of $\alpha^k$ can be given by

$$\alpha^k = \frac{\langle\mathbf{d}^k, \nabla_{\boldsymbol{\theta}}h(\boldsymbol{\theta}^k)\rangle}{(\mathbf{d}^k)^\top\nabla_{\boldsymbol{\theta}}^2 h(\boldsymbol{\theta}^k)\mathbf{d}^k} \tag{6}$$

if $(\mathbf{d}^k)^\top\nabla_{\boldsymbol{\theta}}^2 h(\boldsymbol{\theta}^k)\mathbf{d}^k$ is nonzero. If $\nabla_{\boldsymbol{\theta}}^2 h(\boldsymbol{\theta}^k)$ is positive definite and $\mathbf{d}^k = (\nabla_{\boldsymbol{\theta}}^2 h(\boldsymbol{\theta}^k))^{-1}\nabla_{\boldsymbol{\theta}}h(\boldsymbol{\theta}^k)$, then $\alpha^k = 1$, which leads to the classical Netwon method. In gradient decent methods, the learning rate is often regarded as the most important hyperparameter that highly influences model training. A fixed learning rate may lead to slow convergence or suboptimal performance in some cases. Therefore, many works have considered using adaptive learning rate in gradient decent methods [5, 6, 28]. In the following, we consider giving an adaptive element-wise learning rate automatically scaled by the direction $\mathbf{d}^k$. In this case, the update rule of parameters is given by $\boldsymbol{\theta}^{k+1} = \boldsymbol{\theta}^k - \boldsymbol{\alpha}^k \cdot \mathbf{d}^k = \boldsymbol{\theta}^k - \mathbf{D}^k\boldsymbol{\alpha}^k$, where $\boldsymbol{\alpha}^k \in \mathbb{R}^n$ is the learning rate, $\mathbf{D}^k \in \mathbb{R}^{n \times n}$ is a diagonal matrix with $(\mathbf{D}^k)_{ii} = (\mathbf{d}^k)_i$ and $(\mathbf{D}^k)_{ij} = 0$ when $i \neq j$ for $i, j \in [n]$ and "$\cdot$" denotes the element-wise product. The second Taylor expansion of $h(\boldsymbol{\theta} - \mathbf{D}\boldsymbol{\alpha})$ at iteration $k$ is

$$h(\boldsymbol{\theta}^k - \mathbf{D}^k\boldsymbol{\alpha}^k) \approx h(\boldsymbol{\theta}^k) - \langle\mathbf{D}^k\boldsymbol{\alpha}^k, \nabla_{\boldsymbol{\theta}}h(\boldsymbol{\theta}^k)\rangle + \frac{1}{2}(\mathbf{D}^k\boldsymbol{\alpha}^k)^\top\nabla_{\boldsymbol{\theta}}^2 h(\boldsymbol{\theta}^k)\mathbf{D}^k\boldsymbol{\alpha}^k.$$

Taking the derivative of $h$ with respect to $\boldsymbol{\alpha}^k$ and letting it equal to $\mathbf{0}$, we get

$$2\mathbf{D}^k \nabla_{\boldsymbol{\theta}}^2 h(\boldsymbol{\theta}^k) \mathbf{D}^k \boldsymbol{\alpha}^k - \mathbf{D}^k \nabla_{\boldsymbol{\theta}} h(\boldsymbol{\theta}^k) = \mathbf{0},$$

which yields that

$$\boldsymbol{\alpha}^k = (\nabla_{\boldsymbol{\theta}}^2 h(\boldsymbol{\theta}^k) \mathbf{D}^k)^{-1} \nabla_{\boldsymbol{\theta}} h(\boldsymbol{\theta}^k) \tag{7}$$

if $\mathbf{D}^k$ and $\nabla_{\boldsymbol{\theta}}^2 h(\boldsymbol{\theta}^k)$ are positive definite.

Note that in Eq. (7), it is impractical to compute the inverse of $\nabla_{\boldsymbol{\theta}}^2 h(\boldsymbol{\theta}^k) \mathbf{D}^k$ directly for large-scale models due to high computational and storage costs. For second-order optimization methods in deep learning, some methods have considered approximating the curvature matrix by a block diagonal one according to different layers [10, 14, 15, 19, 24]. What's more, some works have observed that parameters in the same layer have gradients of similar magnitudes. Therefore, a common learning rate can be efficiently shared by these parameters [29, 30]. Inspired by these works, we propose a novel adaptive layer-wise learning method as follows. Suppose that $\mathbf{d}^k = ((\mathbf{d}_1^k)^\top, (\mathbf{d}_2^k)^\top, \ldots, (\mathbf{d}_L^k)^\top)^\top$ is the update direction of a $L$ layers neural network at the iteration point $\boldsymbol{\theta}^k = ((\boldsymbol{\theta}_1^k)^\top, (\boldsymbol{\theta}_2^k)^\top, \ldots, (\boldsymbol{\theta}_L^k)^\top)^\top$, the update rule of $\boldsymbol{\theta}^k$ is given as $\boldsymbol{\theta}^{k+1} = \boldsymbol{\theta}^k - \tilde{\mathbf{D}}^k \tilde{\boldsymbol{\alpha}}^k$, where

$$\tilde{\mathbf{D}}^k = \text{diag}(\mathbf{d}_1^k, \mathbf{d}_2^k, \ldots, \mathbf{d}_L^k) \tag{8}$$

is a block diagonal matrix and $\tilde{\boldsymbol{\alpha}}^k \in \mathbb{R}^L$ is the learning rate. The approximate second Taylor expansion of $h(\boldsymbol{\theta} - \tilde{\mathbf{D}} \tilde{\boldsymbol{\alpha}})$ at iteration $k$ is

$$h(\boldsymbol{\theta}^k - \tilde{\mathbf{D}}^k \tilde{\boldsymbol{\alpha}}^k) \approx h(\boldsymbol{\theta}^k) - \langle \tilde{\mathbf{D}}^k \tilde{\boldsymbol{\alpha}}^k, \nabla_{\boldsymbol{\theta}} h(\boldsymbol{\theta}^k) \rangle + \frac{1}{2} (\tilde{\mathbf{D}}^k \tilde{\boldsymbol{\alpha}}^k)^\top \mathbf{H}^k \tilde{\mathbf{D}}^k \tilde{\boldsymbol{\alpha}}^k, \tag{9}$$

where $\mathbf{H}^k = \text{diag}(\mathbf{H}_1^k, \mathbf{H}_2^k, \ldots, \mathbf{H}_L^k)$ and $\mathbf{H}_l^k = \nabla_{\boldsymbol{\theta}_l}^2 h(\boldsymbol{\theta}^k)$ for $l \in [L]$ and the Hessian matrix is approximated by the block diagonal matrix $\mathbf{H}^k$. Taking the derivative of $h$ with respect to $\tilde{\boldsymbol{\alpha}}^k$ and letting it equal to $\mathbf{0}$, we get $(\tilde{\mathbf{D}}^k)^\top \nabla_{\boldsymbol{\theta}} h(\boldsymbol{\theta}^k) = (\tilde{\mathbf{D}}^k)^\top \mathbf{H}^\top \tilde{\mathbf{D}}^k \tilde{\boldsymbol{\alpha}}^k$, which yields that

$$\boldsymbol{\alpha}^k = \text{diag}(\Theta_1, \Theta_2, \ldots, \Theta_L)^{-1} ((\mathbf{d}_1^k)^\top \nabla_{\boldsymbol{\theta}_1} h(\boldsymbol{\theta}^k), (\mathbf{d}_2^k)^\top \nabla_{\boldsymbol{\theta}_2} h(\boldsymbol{\theta}^k), \ldots (\mathbf{d}_L^k)^\top \nabla_{\boldsymbol{\theta}_L} h(\boldsymbol{\theta}^k))^\top \tag{10}$$

if $\Theta_l$ is nonzero, where $\Theta_l = (\mathbf{d}_1^k)^\top \mathbf{H}_1^k (\mathbf{d}_1^k)$ for $l \in [L]$.

If a same learning rate is used for all layers, as the same way of computing the adaptive layer-wise learning rate, we can get

$$\alpha = \frac{(\mathbf{d}^k)^\top \nabla_{\boldsymbol{\theta}} h(\boldsymbol{\theta}^k)}{(\mathbf{d}^k)^\top \mathbf{H}^k \mathbf{d}^k}. \tag{11}$$

**Theorem 1.** *Let $g(\boldsymbol{\theta})$ and $g_L(\boldsymbol{\theta})$ be the approximate second Taylor expansions of $h(\boldsymbol{\theta} - \alpha \mathbf{d})$ and $h(\boldsymbol{\theta} - \tilde{\mathbf{D}} \tilde{\boldsymbol{\alpha}})$ as given in (9), where $\tilde{\mathbf{D}} \in \mathbb{R}^{n \times L}$, $\tilde{\boldsymbol{\alpha}} \in \mathbb{R}^L$ and $\alpha \in \mathbb{R}$ are given by (8), (10) and (11) respectively, then we have $g_L(\boldsymbol{\theta}) \leq g(\boldsymbol{\theta})$.*

*Proof.* The proof is given in the appendix. □

By Theorem 1, we know that the adaptive layer-wise learning rate may lead to a faster decline in terms of function values. In our proposed algorithm, we choose $\mathbf{d}^k = (\mathbf{F}^k)^{-1} \nabla_{\boldsymbol{\theta}} h(\boldsymbol{\theta}^k)$, where $\mathbf{F}^k$ is the Fisher information matrix and can be seen as a approximation of the Hessian matrix. Then, the Fisher information matrix is approximated by a block diagonal matrix each block matrix is approximated by the Kronecker product of two factor matrices. In each layer, the update direction $\mathbf{d}_l^k$ is scaled by a layer-wise damping learning rate $\alpha_l^k$ according to (10), which is given by

$$\alpha_l^k = \frac{(\mathbf{d}_l^k)^\top \nabla_{\boldsymbol{\theta}} h(\boldsymbol{\theta}_l^k)}{(\mathbf{d}_l^k)^\top \mathbf{F}_l^k \mathbf{d}_l^k + \mu}, \tag{12}$$

where $\mu > 0$ is a parameter. Using this adaptive layer-wise learning rate can accelerate layers with smaller gradients. Moreover, this approach can also avoid computing the inverse matrix in element-wise learning rate (13) and remain computationally efficient.

## 3.3 Algorithm Schema

To effectively apply LNGD in training DNNs, several certain techniques need to be employed. In this section, we primarily focus on introducing the damping technique, which is a commonly used in second-order methods. Meanwhile, a simple method can be used to compute the adaptive layer-wise learning rate according to Eq. (12) since the cost of computing $(\mathbf{d}_l^k)^\top \mathbf{F}_l^k \mathbf{d}_l^k$ is relatively expensive. Finally, we discuss the utilization of exponential moving averages to enhance the training process.

**A new damping technique**: Damping plays a crucial role in second-order optimization methods. Large damping can weaken the effect of curvature matrix, while small damping may cause computational difficulty and inaccuracy since most eigenvalues of the Fisher information matrix are close to zero and only a small number of eigenvalues take on large values. To make training stable, we propose the following damping for the $l$th layer: $\lambda_l = \min(\max(\mathrm{tr}(\mathbf{F}_l)/d_l, \nu_1), \nu_2)$, where $\nu_1$ and $\nu_2$ are two constants to constrain the minimum and maximum of damping, and $d_l$ is the number of weight parameters. In our method, $\mathbf{F}_l$ is approximated as the Kronecker product of two factors $\boldsymbol{\Phi}_l$ and $\boldsymbol{\Psi}_l$, so we add the damping to each factors by $\hat{\boldsymbol{\Phi}}_l = \boldsymbol{\Phi}_l + \lambda_l^\Phi$ and $\hat{\boldsymbol{\Psi}}_l = \boldsymbol{\Psi}_l + \lambda_l^\Psi$, where $\lambda_l^\Phi = \min(\max(\mathrm{tr}(\boldsymbol{\Phi}_l)/n, \nu_1), \nu_2)$ and $\lambda_l^\Psi = \min(\max(\mathrm{tr}(\boldsymbol{\Psi}_l)/n, \nu_1), \nu_2)$.

**Compute the learning rate the faster**: In order to compute the adaptive layer-wise learning rate given in Eq. (12) more quickly, we turn matrix computation into vector computation. Specifically,

$$
\begin{aligned}
(\mathbf{d}_l^k)^\top \mathbf{F}_l^k \mathbf{d}_l^k = (\mathbf{d}_l^k)^\top \mathop{\mathbb{E}}_{(x,y\sim p(x,y))}[\mathcal{D}\boldsymbol{\theta}_l \mathcal{D}\boldsymbol{\theta}_l^\top]\mathbf{d}_l^k &= \mathop{\mathbb{E}}_{(x,y\sim p(x,y))}[(\mathbf{d}_l^k)^\top \mathcal{D}\boldsymbol{\theta}_l \mathcal{D}\boldsymbol{\theta}_l^\top \mathbf{d}_l^k] \\
&= \mathop{\mathbb{E}}_{(x,y\sim p(x,y))}[((\mathbf{d}_l^k)^\top \mathcal{D}\boldsymbol{\theta}_l)^2] \approx \frac{1}{N}[((\mathbf{d}_l^k)^\top \mathcal{D}\boldsymbol{\theta}_l)^2],
\end{aligned}
\tag{13}
$$

where $\mathbf{F}_l^k$ is the empirical Fisher information matrix and $N$ is the number of samples. The empirical version of Fisher information matrix with no need for sampling from the model's prediction distribution, making it more computationally efficient.

**Exponential moving averages**: In line with previous studies, we incorporate exponential moving averages into our approach. This involves updating the estimate by combining the previous estimate, weighted by $\epsilon$, with the estimate calculated from the new mini-batch, weighted by $1 - \epsilon$. That is

$$
\hat{\boldsymbol{\Phi}}_l^{k+1} \leftarrow \epsilon\hat{\boldsymbol{\Phi}}_l^{k+1} + (1-\epsilon)\hat{\boldsymbol{\Phi}}^k \text{ and } \hat{\boldsymbol{\Psi}}_l^{k+1} \leftarrow \epsilon\hat{\boldsymbol{\Psi}}_l^{k+1} + (1-\epsilon)\hat{\boldsymbol{\Psi}}_l^k.
\tag{14}
$$

In summary, our proposed algorithm is shown in Algorithm1.

## 3.4 Convergence Analysis

In this subsection, we give the convergence analysis of LNGD. Following the model used in previous works about analysing the gradient descent [31, 32, 33] and NGD [34, 35], we consider a two-layer neural network activated by the ReLU function with $m$ neurons in the hidden layer as follows:

$$
f(\boldsymbol{\theta}, a, \mathbf{x}) = \frac{1}{\sqrt{m}} \sum_{r=1}^{m} a_r \varphi(\boldsymbol{\theta}_r^\top \mathbf{x}),
$$

where $\boldsymbol{\theta}_1, \boldsymbol{\theta}_2, \ldots, \boldsymbol{\theta}_m \in \mathbb{R}^d$ are the weight vectors of the first layer, $\mathbf{x} \in \mathbb{R}^d$ is the input, $a_r \in \mathbb{R}$ is the weight of unit $r$ in the second layer and $\varphi(\cdot)$ is the ReLU activation function, i.e., $\varphi(x) = \max\{0, x\}$. Let $\mathbf{v} = [f(\boldsymbol{\theta}, a, \mathbf{x}_i), f(\boldsymbol{\theta}, a, \mathbf{x}_2), \ldots, f(\boldsymbol{\theta}, a, \mathbf{x}_n)]^\top$. In the following, we only give the result of convergence of Algorithm 1, the specific proof, which uses some conclusions in [36, 37, 38, 39], is given in the appendix.

**Theorem 2.** *(Convergence rate of LNGD) Under the Assumption 1 and the assumption that* $rank(\mathbf{X}) = d$. *If we set the number of hidden units* $m = \Omega\left(\frac{n^4 \kappa_{\mathbf{Z}_{\mathbf{X},\mathbf{G}}}^8}{\nu^2 \varepsilon^3 \lambda_{\mathbf{G}}^4}\right)$, *we i.i.d initialize* $\boldsymbol{\theta}_r \sim \mathcal{N}(0, \nu\mathbf{I})$, $a_r \sim \mathrm{unif}[\{-1, +1\}]$ *for any* $r \in [m]$, *and we set the step size* $\alpha \leq \frac{(1-2c)}{(1+c)^2}$. *Then with probability at least* $1 - \varepsilon$ *over the random initialization, we have for* $k = 0, 1, 2, \ldots$

$$
\|\mathbf{y} - \mathbf{v}^k\|_2^2 \leq (1 - \alpha)^k \|\mathbf{y} - \mathbf{v}^0\|_2^2.
$$

*Proof.* The proof is given in the appendix. $\square$

**Algorithm 1** LNGD

---

**Require:** learning rate $\alpha$, learning rate parameter $\mu$, damping parameter $\lambda$, damping constraints $\nu_1, \nu_2$, momentum parameter $\tau$, exponential moving average parameter $\epsilon$, Fisher information matrix and its inverse update intervals $T_{\text{FIM}}$ and $T_{\text{INV}}$.

1: $k \leftarrow 0, \boldsymbol{m} \leftarrow \boldsymbol{0}$. Initialize $\hat{\boldsymbol{\Phi}}_l$ and $\hat{\boldsymbol{\Psi}}_l$ for any $l \in [L]$.
2: **while** convergence is not reached **do**
3:     Select a new mini-batch
4:     **for all** $l \in [L]$ **do**
5:       **if** $k \equiv 0 \pmod{T_{\text{FIM}}}$ **then**
6:         Update the factors $\hat{\boldsymbol{\Phi}}_l$ and $\hat{\boldsymbol{\Psi}}_l$ using Eq. (14)
7:       **end if**
8:       **if** $k \equiv 0 \pmod{T_{\text{INV}}}$ **then**
9:         Compute the inverses of $\hat{\boldsymbol{\Phi}}_l$ and $\hat{\boldsymbol{\Psi}}_l$
10:       **end if**
11:     Compute $\nabla_{\boldsymbol{\theta}_l} h(\boldsymbol{\theta})$ using backpropagation
12:     Compute the approximated natural gradient $(\hat{\boldsymbol{\Phi}}_l^{-1} \otimes \hat{\boldsymbol{\Psi}}_l^{-1}) \nabla_{\boldsymbol{\theta}_l} h(\boldsymbol{\theta})$
13:     Compute the adaptive learning rate $\alpha_l$ using Eq. (12)
14:     $\boldsymbol{\zeta} \leftarrow -\alpha\alpha_l(\hat{\boldsymbol{\Phi}}_l^{-1} \otimes \hat{\boldsymbol{\Psi}}_l^{-1}) \nabla_{\boldsymbol{\theta}_l} h(\boldsymbol{\theta})$
15:     $\boldsymbol{m} \leftarrow \tau\boldsymbol{m} + \boldsymbol{\zeta}$ (Update momentum)
16:     $\boldsymbol{\theta}_l \leftarrow \boldsymbol{\theta}_l + \boldsymbol{m}$ (Update parameters)
17:     **end for**
18:     $k \leftarrow k + 1$
19: **end while**
20: **return** $\boldsymbol{\theta}$

---

# 4 Experiments

In order to verify the effectiveness of the proposed optimizer, we apply the optimizer to both image classification and machine translation tasks. We first present the optimization performance of our optimizer by comparing with several baselines. Then, we pay attention to the contribution of different modules of our optimizer by conducting elaborate ablation analysis, which is given in the appendix. Unless otherwise stated, the batch size for all experiments in the following is set to 256. The initial learning rate hyperparameters for all optimizers are tuned using a grid search with values $\alpha \in \{1e-4, 3e-4, \ldots, 1, 3\}$. The damping parameter $\lambda$ in KFAC[14] are tuned using a grid search with values $\lambda \in \{1e-6, 1e-4, 3e-4, 1e-3, \ldots, 1e-1, 3e-1\}$. The minimum and maximum of damping parameters $\nu_1$ and $\nu_2$ in LNGD are set to $1e-5$ and $1e-2$. The moving average parameter and the momentum correlating with KFAC and LNGD are set to 0.95 and 0.9, respectively. Furthermore, a weight decay of 0.004 is applied in all optimizers. All experiments run on a single A100 GPU using TensorFlow. We average the results of 5 runs and the hyper-parameter settings for these optimizers are the best values randomly searched for many times.

## 4.1 CIFAR-10 Training

We first report the optimizing performance on CIFAR-10 [40], which is a standard task used to benchmark optimization methods [6, 41, 42, 43, 44]. Following these previous works, the changes of testing accuracy and training loss versus time as well as epoch are reported in Fig. 2, and detailed statistics are shown in Table6. From Fig. 2, it can be observed that LNGD exhibits the most rapid decline in training loss during the initial epochs and seconds. This suggests that LNGD is effective in quickly reducing the training loss and reaching convergence. All optimization methods convergent at around 200 epochs. However, it is observed that second-order optimization methods, such as KFAC and LNGD, achieve a lower training loss compared to first-order optimization methods like SGD and Adam. In terms of testing accuracy, as depicted in Fig. 2 (b) and (d), LNGD achieves a top-1 accuracy of 91% at the fastest rate. It only requires 36 epochs and 189.69 seconds to achieve this accuracy level. In comparison, as presented in Table6, SGD and ADAM require at least 100% and 30% more epochs and time, respectively, to achieve similar accuracy. Relative to KFAC, LNGD reduces the number of epochs and time by around 20% and 21%, respectively. Furthermore, as shown in Table6, LNGD gets the highest final testing accuracy after convergence.

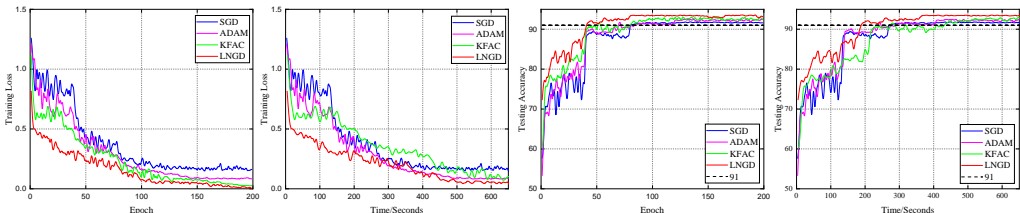

Figure 2: Numerical performance on ResNet-18 with CIFAR-10.

Table 1: Detailed statistics on CIFAR-10 when top-1 testing accuracy achieves 91%.

|        | Epoch | Total Time | Time Per Epoch | Acceleration | Best Test Acc |
|--------|-------|-----------|----------------|--------------|---------------|
| SGD    | 79    | 268.67s   | 3.4s           | 29%          | 91.88%        |
| ADAM   | 72    | 248.83s   | 3.77s          | 23%          | 92.62%        |
| KFAC   | 45    | 241.86s   | 5.87s          | 21%          | 93.34%        |
| LNGD   | 36    | 189.69s   | 5.08s          |              | 93.61%        |

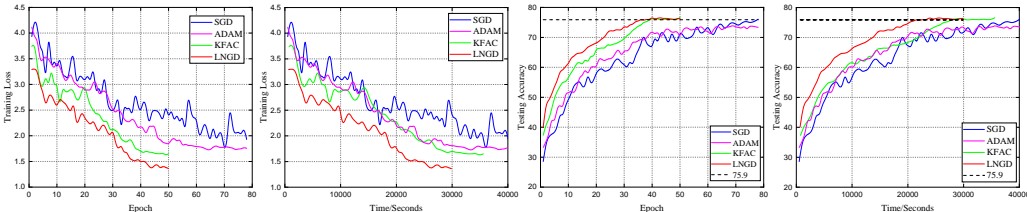

Figure 3: Numerical performance on ResNet-50 with ImageNet.

## 4.2 ImageNet Training

We extend our examination of optimizer efficacy to a larger image classification dataset, ImageNet-1K [45]. The changes of testing accuracy and training loss versus time and epoch are reported in Fig.3 and Table2. The results show that the LNGD optimizer is highly efficient in training large image datasets in terms of both speed and accuracy. LNGD, which requires only 36 epochs and 6.46 hours, is much faster in achieving the top-1 testing accuracy of 75.9% than other baselines. This is a significant improvement over SGD, which takes 100% more epochs and 75% more time to reach the same accuracy level. As for Adam, it exhibits a rapid decrease in loss during training and reaches convergence at a fast rate. However, the best achieved testing accuracy is only 74.05%, indicating that when training large-scale image tasks, a trade-off between efficiency and effectiveness needs to be considered. Compared to KFAC, although LNGD is better for only 3 epochs, it leads to 19% reduction in terms of the computing time. The training loss results further support the efficiency of LNGD, as it maintains the fastest rate of decline during the initial stages of training and ultimately yields the lowest training loss upon convergence. Overall, the results suggest that LNGD is a highly efficient optimizer for large-scale image classification tasks, providing faster convergence and better accuracy than other commonly used optimizers.

## 4.3 Transformer Training

In this experiment, we apply LNGD to the Transformer-Big model [46] with 213.7M parameters. The training datasets is WMT English-German machine translation corpus [46]. We use Bleu [47] as the evaluation metrics, which is frequently used in machine translation tasks. The setting of learning rate updating strategy for SGD, Adam, KFAC and LNGD are the same as in ImageNet training.

In Fig.4 and Table3, we present the comparative evaluation of the performance of LNGD against SGD, Adam, and KFAC in terms of testing accuracy and training loss. ADAM demonstrates superior performance over SGD, as evidenced by a more rapid decrease in training loss and a lower converged loss value. This observation aligns with previous empirical findings that ADAM is highly effective for transformer models. KFAC exhibits further enhancements in performance compared to Adam,

Table 2: Detailed statistics on ImageNet when top-1 testing accuracy achieves 75.9%.

|       | Epoch | Total Time | Time Per Epoch | Acceleration | Best Test Acc |
|-------|-------|------------|----------------|--------------|---------------|
| SGD   | 78    | 11.28h     | 520.55s        | 43%          | 76.47%        |
| ADAM  | -     | -          | -              | -            | 74.05%        |
| KFAC  | 39    | 8.02h      | 739.93s        | 19%          | 76.58%        |
| LNGD  | 36    | 6.46h      | 646.44s        |              | 76.73%        |

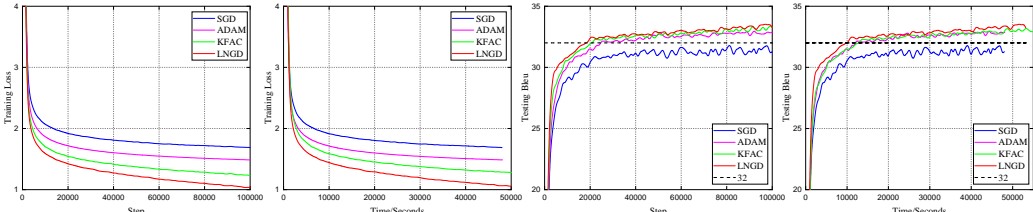

Figure 4: Numerical performance on Transformer with WMT.

yet it does not surpass the efficacy of LNGD. LNGD outperforms its counterparts with the swiftest reduction in training loss and the highest convergence rates. In terms of testing accuracy, measured by the Bleu score, LNGD achieves a top-1 Bleu score of 32% with remarkable efficiency, which is able to reduce the required steps by approximately 24% and computing time by 16% compared to Adam. When compared to KFAC, LNGD still shows significant improvements, reducing the steps by around 14% and computing time by 24%. As for SGD, it cannot reach the top-1 Bleu score of 32% and the best testing accuracy is only 31.8%, which indicates that SGD is not a good choice for large language processing tasks. In summary, the results provide strong evidences for the effectiveness of LNGD as an optimization algorithm for transformer models and shed light for large practical NLP tasks where time and computational resources are quite limited.

Table 3: Detailed statistics on WMT when Bleu achieves 32%.

|       | Step | Total Time | Time Per 1K | Acceleration | Best Test Bleu |
|-------|------|------------|-------------|--------------|----------------|
| SGD   | -    | -          | -           | -            | 31.87%         |
| ADAM  | 25K  | 3.39h      | 488.16s     | 16%          | 33.05%         |
| KFAC  | 22K  | 3.75h      | 613.63s     | 24%          | 33.45%         |
| LNGD  | 19K  | 2.85h      | 540s        |              | 33.55%         |

## 5 Conclusion

In summary, we propose a novel NGD optimizer named as LNGD for training DNNs, specifically targeting the computational inefficiencies that impede the practical application of conventional natural gradient techniques in large-scale neural networks. Our approach strategically computes Fisher information matrices for each individual layers using sample approximation and dynamically adjusts learning rates leveraging curvature information. This method facilitates a more refined representation of the optimization landscape at the layer level. Besides, we provide convergence analysis of LNGD. Experimental evaluations indicate its competitive performance relative to existing state-of-the-art optimizers. This work hold significant potential for enhancing the efficiency and scalability of training processes in deep learning frameworks.

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

## A  Notations

In this paper, we denote $[n] = \{1, 2, \ldots, n\}$. For a matrix $\mathbf{A}$, we use $\mathbf{A}_{ij}$ to denote its $(i, j)$-th entry, $\text{tr}(\mathbf{A})$ to denote its trace, $\sigma_{\min}(\mathbf{A})$ and $\sigma_{\max}(\mathbf{A})$ to denote its smallest and largest singular value, $\|\mathbf{A}\|_{\mathbf{F}}$ to denote its Frobenius norm and $\|\mathbf{A}\|_2$ to denote its spectral norm. If $\mathbf{A}$ is positive semi-definite, $\lambda_{\min}(\mathbf{A})$ and $\lambda_{\max}(\mathbf{A})$ denote its smallest and largest eigenvalue, and define $\kappa_{\mathbf{A}} = \lambda_{\max}(\mathbf{A})/\lambda_{\min}(\mathbf{A})$, respectively. The identity matrix is denoted as $\mathbf{I}$. For a vector $\mathbf{a}$, $\|\mathbf{a}\|_2$ denotes the Euclidean norm. We use $\mathcal{N}(\boldsymbol{\mu}, \boldsymbol{\Sigma})$ to denote the Gaussian distribution with mean $\boldsymbol{\mu}$ and covariance $\boldsymbol{\Sigma}$. For two matrices

$$\mathbf{A} = [\mathbf{a}_1, \mathbf{a}_2, \ldots, \mathbf{a}_n] \in \mathbb{R}^{p \times q}, \quad \mathbf{B} = [\mathbf{b}_1, \mathbf{b}_2, \ldots, \mathbf{b}_n] \in \mathbb{R}^{p \times q},$$

we use $\circ$ and $\otimes$ to denote the Hadamard and Kronecker product, respectively. The column-wise Khatri-Rao product $*$ is defines as

$$\mathbf{A} * \mathbf{B} = [\mathbf{a}_1 \otimes \mathbf{b}_1, \mathbf{a}_2 \otimes \mathbf{b}_2, \ldots, \mathbf{a}_n \otimes \mathbf{b}_n] \in \mathbb{R}^{p^2 \times q}.$$

Similarly, we can define the row-wise Khatri-Rao product $\star$ and we have $(\mathbf{A} \star \mathbf{B})^\top = (\mathbf{A}^\top * \mathbf{B}^\top)$. Given an event $E$, $\mathbb{I}\{E\}$ denotes its indicator function, i.e.,

$$\mathbb{I}\{E\} = \begin{cases} 1, & \text{if } E \text{ happens,} \\ 0, & \text{otherwise.} \end{cases}$$

## B  Comparisons and Explanations

### B.1  Comparisons with Related Works

There have been some works to utilize the NGD or its approximations for training DNNs. One of the primary computational challenges lie in the storage and computing the inverse of the Fisher information matrix during NGD optimization. Recently, several studies have explored the adoption of the efficient Kronecker-factored approximation to the Fisher information matrix to address this computational challenge. The most related approaches to this work are the KFAC, EKFAC and TKFAC[15]. These works and LNGD all start with a block-diagonal approximation of the Fisher information matrix. The differences among them are the approximations of the block matrix $\mathbf{F}_l$. By approximating the expectation of the Kronecker product as the Kronecker product of expectations, KFAC approximates $\mathbf{F}_l$ as the Kronecker product of $\mathbf{A} = \mathbb{E}[(\mathbf{a}_{l-1}\mathbf{a}_{l-1}^\top)]$ and $\mathbf{B} = \mathbb{E}[(\hat{\mathbf{g}}_l\hat{\mathbf{g}}_l^\top)]$ with $\hat{\mathbf{g}}_l = -\nabla_{\mathbf{s}_l} \log p(\mathbf{y}|\mathbf{x})$. By tracking the diagonal variance in the Kronecker-factored eigenbasis, EKFAC performs eigenvalue decomposition of the Fisher information matrix and re-scales the eigenvalues by $\mathbf{S}^*$ to achieve a better approximation, where $\mathbf{S}^*$ is a diagonal matrix defined by $\mathbf{S}_{ii}^* = \mathbb{E}[(\mathbf{U}_{\mathbf{B}}^\top \nabla_{\boldsymbol{\theta}} h(\boldsymbol{\theta})^2)_i]$, and $\mathbf{U}_{\mathbf{A}}, \mathbf{U}_{\mathbf{B}}$ are eigenvectors of $\mathbf{A}, \mathbf{B}$. TKFAC approximates $\mathbf{F}_l$ as a Kronecker product of two factors $\mathbf{P}$ and $\mathbf{Q}$ scaled by a coefficient $\delta$ and keep the traces of each block equal. In this paper, we propose the LNGD, which approximates $\mathbf{F}_l$ as a Kronecker product of a matrix $\boldsymbol{\Phi}_l$ and a diagonal matrix $\boldsymbol{\Psi}_l$, which is computed by sampling from each layer. We summarize the above approximations of $\mathbf{F}_l$ in Table 4.

Table 4: Summary of some NGD optimizers

| Optimizer | $\mathbf{F}_l$ |
|---|---|
| KFAC | $\mathbf{A} \otimes \mathbf{B}$ |
| EKFAC | $(\mathbf{U}_{\mathbf{A}} \otimes \mathbf{U}_{\mathbf{B}})\mathbf{S}^*(\mathbf{U}_{\mathbf{A}} \otimes \mathbf{U}_{\mathbf{B}})^\top$ |
| TKFAC | $\delta\mathbf{P} \otimes \mathbf{Q}$ |
| LNGD | $\boldsymbol{\Phi} \otimes \boldsymbol{\Psi}$ |

When these methods have the same update frequency, KFAC needs to compute two factor matrices $\mathbf{A}$ and $\mathbf{B}$, and then invert them. However, $\mathbf{B}$ can only be computed after completely performing back-propagation. On the other hand, EKFAC modifies KFAC by incorporating eigenvalue decomposition to scale the eigenvalue during the inversion process. TKFAC, another variant of KFAC, maintains the equality of traces of matrices before and after approximation. Both EKFAC and TKFAC involve increased computational requirements compared to KFAC. Our proposed LNGD also requires computation and inversion of two factors $\boldsymbol{\Phi}$ and $\boldsymbol{\Psi}$. However, the advantage of LNGD is that the

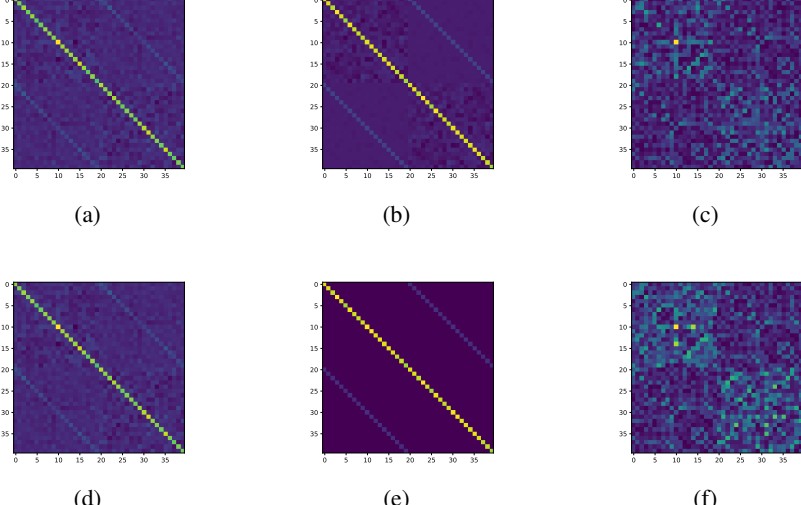

Figure 5: Comparison of the exact Fisher information matrix and the approximated Fisher information matrix of KFAC and LNGD. On the left is the exact Fisher information matrix, in the middle is the approximated Fisher information matrix, and on the right is the absolute error of these. The first row shows the result of KFAC, and the second row shows the results of LNGD.

matrix $\mathbf{\Psi}$ is diagonal, and due to the use of hierarchical sampling, $\mathbf{\Psi}$ can be directly inverse during the forward propagation phase, without the need to wait for the completion of the back-propagation process. This feature significantly reduces computational time.

In addition, THOR proposed in [24] also provides an optimizer using NGD for training DNNs. THOR mainly considers reducing the computational cost of NGD through two aspects. On the one hand, THOR gradually increases the updating frequency of the inverse matrix of Fisher information matrix and proposes a trace-based updating rule for the Fisher information matrix of each layer. On the other hand, THOR approximates the approximated Fisher information matrix obtained by KFAC as some smaller matrices by splitting matrix dimensions. Our proposed LNGD first gives a layer-wise sample method to more efficiently compute each block matrix corresponding to each layer and proposes a novel approximate scheme of the Fisher information matrix. Furthermore, LNGD also adopts an adaptive layer-wise learning rate to speed up training. The contributions and ideas of our proposed LNGD are different from THOR.

### B.2 Comparisons Between KFAC and LNGD

Fig. 5 shows the visualization results of KFAC and LNGD. From Fig. 5 (b) and (e), we can see that KFAC and LNGD can all emphasize the importance of the diagonal elements in the exact Fisher information matrix. In addition, it can also be seen clearly that KFAC still retains some elements near the main diagonal, while LNGD does not, which also reflects that LNGD provides an efficient approximation of Fisher information matrix with less computational cost in comparison with KFAC.

### B.3 Illustration of the Gaussian Distribution Assumption

In Subsection 3.1, we assume that the predictive distribution $P_{\mathbf{a}_l|\mathbf{a}_{l-1}}(\boldsymbol{\theta}_l)$ follows Gaussian distribution. To illustrate the validity of the Gaussian distribution assumption, we collect the output of two layers of the ResNet-18 network on CIFAR-10 and show the results in Fig. 6. Fig. 6 (a) and (b) show the distributions of sample representation vectors' values in some dimension. Since we use the ReLU activation function, the obtained distributions are in accord with the Gaussian distribution in the positive quadrant. Fig. 6 (c) and (d) show the distributions of values of sample representation vectors' Euclidean norm, from which we can see that the two distributions can also be approximated as Gaussian distributions.

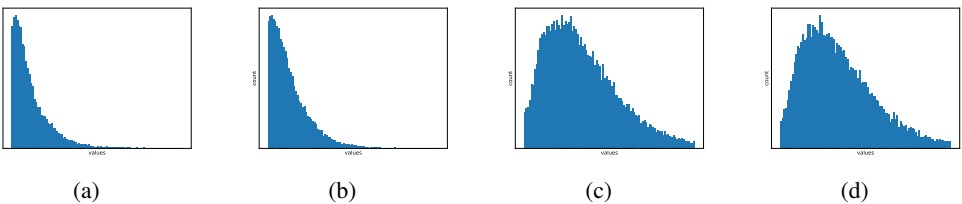

$$\text{(a)} \qquad\qquad \text{(b)} \qquad\qquad \text{(c)} \qquad\qquad \text{(d)}$$

Figure 6: Illustration of Gaussian distribution.

## C Proof of Theorem 1

*Proof.* Since $g(\boldsymbol{\theta})$ and $g_L(\boldsymbol{\theta})$ are the approximate second Taylor expansions of $h(\boldsymbol{\theta} - \alpha\mathbf{d})$ and $h(\boldsymbol{\theta} - \tilde{\mathbf{D}}\tilde{\boldsymbol{\alpha}})$, we have

$$g_L(\boldsymbol{\theta}) - g(\boldsymbol{\theta}) = h(\boldsymbol{\theta}) - \alpha\langle\mathbf{d}, \nabla_{\boldsymbol{\theta}} h(\boldsymbol{\theta})\rangle + \frac{1}{2}\alpha^2\mathbf{d}^\top\mathbf{H}\mathbf{d} - h(\boldsymbol{\theta}) + \sum_{l=1}^{L}\tilde{\alpha}_l\langle\mathbf{d}_l, \nabla_{\boldsymbol{\theta}_l} h(\boldsymbol{\theta})\rangle$$

$$-\sum_{l=1}^{L}\frac{1}{2}\tilde{\alpha}_l^2\mathbf{d}_l^\top\mathbf{H}_l\mathbf{d}_l = \frac{1}{2}\left(\sum_{l=1}^{L}\frac{(\mathbf{d}_l^\top\nabla_{\boldsymbol{\theta}_l}h(\boldsymbol{\theta}))^2}{\mathbf{d}_l^\top\mathbf{H}_l\mathbf{d}_l} - \frac{(\mathbf{d}^\top\nabla_{\boldsymbol{\theta}}h(\boldsymbol{\theta}))^2}{\mathbf{d}^\top\mathbf{H}\mathbf{d}}\right) \le 0.$$

This completes the proof. $\qquad\qquad\qquad\qquad\qquad\qquad\qquad\qquad\qquad\qquad\qquad\qquad\qquad\qquad\square$

## D Convergence of LNGD

In this section, we give the convergence analysis of LNGD. Following the model used in previous works about analysing the gradient descent [31, 32, 33] and NGD [34, 35], we consider a two-layer neural network activated by the ReLU function with $m$ neurons in the hidden layer as follows:

$$f(\boldsymbol{\theta}, a, \mathbf{x}) = \frac{1}{\sqrt{m}}\sum_{r=1}^{m}a_r\varphi(\boldsymbol{\theta}_r^\top\mathbf{x}),$$

where $\boldsymbol{\theta}_1, \boldsymbol{\theta}_2, \ldots, \boldsymbol{\theta}_m \in \mathbb{R}^d$ are the weight vectors of the first layer, $\mathbf{x} \in \mathbb{R}^d$ is the input, $a_r \in \mathbb{R}$ is the weight of unit $r$ in the second layer and $\varphi(\cdot)$ is the ReLU activation function, i.e., $\varphi(x) = \max\{0, x\}$. For convenience, we define $\boldsymbol{\theta} = [\boldsymbol{\theta}_1^\top, \boldsymbol{\theta}_2^\top, \ldots, \boldsymbol{\theta}_m^\top]^\top \in \mathbb{R}^{md}$. We first initialize the parameters randomly by

$$\boldsymbol{\theta}_r \sim \mathcal{N}(\mathbf{0}, \nu^2\mathbf{I}), \quad a_r \sim \text{unif}[\{-1, +1\}], \quad \forall r \in [m],$$

where $0 < \nu \le 1$ controls the magnitude of initialization.

Given the training dataset $\mathcal{S} = \{(\mathbf{x}_i, y_i)\}_{i=1}^{n}$ containing (input, target) examples $(\mathbf{x}_i, y_i)$. Following [31, 33, 34], we make the following assumption for the data.

**Assumption 1.** *For all $i$, $\|\mathbf{x}_i\|_2^2 = 1$ and $|y_i| = \mathcal{O}(1)$. For any $i \ne j$, $\mathbf{x}_i \nparallel \mathbf{x}_j$.*

In this subsection, we mainly focus on the mean squared error loss (MSE) function

$$\mathcal{L}(\boldsymbol{\theta}) = \frac{1}{2n}\sum_{i=1}^{n}(f(\boldsymbol{\theta}, a, \mathbf{x}_i) - y_i)^2$$

$$= \frac{1}{2n}\sum_{i=1}^{n}(\frac{1}{\sqrt{m}}\sum_{r=1}^{m}a_r\varphi(\boldsymbol{\theta}_r^\top\mathbf{x}) - y_i)^2.$$

Following [31, 33, 34], we fix the weights of second layer and only optimize the weights of first layer. Then the update rule of NGD can be written as

$$\boldsymbol{\theta}^{k+1} = \boldsymbol{\theta}^k - \alpha(\mathbf{F}^k)^{-1}\nabla_{\boldsymbol{\theta}}\mathcal{L}(\boldsymbol{\theta}^k).$$

As shown in [18], if the network's predictive distribution is in the exponential family, the Fisher information matrix is equivalent to the generalized Gauss-Newton matrix, which is defined by

$$\mathbb{E}_{(\mathbf{x}_i,y_i)\in\mathcal{S}}[\mathbf{J}_i^\top\mathbf{H}_\mathcal{L}\mathbf{J}_i],$$

where $\mathbf{H}_\mathcal{L}$ is the Hessian matrix of the loss function $\mathcal{L}(\boldsymbol{\theta})$ with respect to the prediction $f(\boldsymbol{\theta},a,\mathbf{x}_i)$ and $\mathbf{J}_i$ is the Jacobian matrix of $f(\boldsymbol{\theta},a,\mathbf{x}_i)$ with respect to the parameters $\boldsymbol{\theta}$. Under our setting that $\mathcal{L}(\boldsymbol{\theta})$ is the MSE loss function, the Hessian matrix $\mathbf{H}_\mathcal{L}$ is the identity matrix $\mathbf{I}$. $\mathbf{J}_i$ can be computed by

$$\mathbf{J}_i = \left(\nabla_{\boldsymbol{\theta}_1}f(\boldsymbol{\theta},a,\mathbf{x}_i)^\top,\ldots,\nabla_{\boldsymbol{\theta}_m}f(\boldsymbol{\theta},a,\mathbf{x}_i)^\top\right)^\top,$$

where

$$\nabla_{\boldsymbol{\theta}_r}f(\boldsymbol{\theta},a,\mathbf{x}_i) = \frac{a_r}{\sqrt{m}}\mathbb{I}\{\boldsymbol{\theta}_r^\top\mathbf{x}_i\geq 0\}\mathbf{x}_i, \quad \forall\, r\in[m]. \tag{15}$$

Let $\mathbf{J} = (\mathbf{J}_1,\mathbf{J}_2,\ldots,\mathbf{J}_n)^\top \in \mathbb{R}^{n\times md}$, then the Fisher information matrix can be written as

$$\mathbf{F} = \mathbb{E}_{(\mathbf{x}_i,y_i)\in\mathcal{S}}[\mathbf{J}_i^\top\mathbf{H}_\mathcal{L}\mathbf{J}_i] = \mathbb{E}_{(\mathbf{x}_i,y_i)\in\mathcal{S}}[\mathbf{J}_i^\top\mathbf{J}_i] = \frac{1}{n}\mathbf{J}^\top\mathbf{J}.$$

As discussed in [34], when $m > n$, the Fisher information matrix is singular. So in this case, we use the generalized inverse given in [36]

$$\mathbf{F}^\dagger = n\mathbf{J}^\top(\mathbf{J}\mathbf{J}^\top)^{-1}(\mathbf{J}\mathbf{J}^\top)^{-1}\mathbf{J} \tag{16}$$

and the update rule of NGD can be written as

$$\boldsymbol{\theta}^{k+1} = \boldsymbol{\theta}^k - \frac{\alpha}{n}(\mathbf{F}^k)^\dagger(\mathbf{J}^k)^\top(\mathbf{v}^k - \mathbf{y}),$$

where $\mathbf{y} = [y_1,y_2,\ldots,y_n]^\top$ and $\mathbf{v} = [v_1,v_2,\ldots,v_n]^\top = [f(\boldsymbol{\theta},a,\mathbf{x}_i),f(\boldsymbol{\theta},a,\mathbf{x}_2),\ldots,f(\boldsymbol{\theta},a,\mathbf{x}_n)]^\top$. Consider the two-layer neural network described in this subsection, since we fixed the weights in second layer, and the Fisher information matrix of this model is approximated by

$$\mathbf{F} \approx \boldsymbol{\Phi} \otimes \boldsymbol{\Psi}.$$

For simplicity, we ignore the index of layer. Define

$$\mathbf{X} = [\mathbf{x}_1,\mathbf{x}_2,\ldots,\mathbf{x}_n]^\top \in \mathbb{R}^{n\times d}$$

and

$$\mathbf{G} = [\varphi'(\mathbf{X}\boldsymbol{\theta}_1),\varphi'(\mathbf{X}\boldsymbol{\theta}_2),\ldots,\varphi'(\mathbf{X}\boldsymbol{\theta}_m)] \in \mathbb{R}^{n\times m},$$

where $\mathbf{X}$ is the input matrix formed by $n$ input vectors and $\mathbf{G}$ is the pre-activation derivatives matrix whose entry is given by

$$\mathbf{G}_{ir} = \frac{1}{\sqrt{m}}\mathbb{I}\{\mathbf{x}_i^\top\boldsymbol{\theta}_r \geq 0\}, \quad \forall\, i\in[n], r\in[m]. \tag{17}$$

Then we have

$$\mathbf{J} = \nabla_{\boldsymbol{\theta}}f(\boldsymbol{\theta},a,\mathbf{x}_i) = \mathbf{X}\star\mathbf{G} \in \mathbb{R}^{n\times md}$$

and

$$\mathbf{F} = \frac{1}{n}\mathbf{J}^\top\mathbf{J} = \frac{1}{n}(\mathbf{X}\star\mathbf{G})^\top(\mathbf{X}\star\mathbf{G}) \in \mathbb{R}^{md\times md}.$$

The formulas of $\boldsymbol{\Phi}$ and $\boldsymbol{\Psi}$ can be given as

$$\boldsymbol{\Phi} = \mathbf{X}^\top(\mathbf{G}\mathbf{G}^\top\circ\mathbf{I})\mathbf{X} \in \mathbb{R}^{d\times d}, \tag{18}$$

$$\boldsymbol{\Psi} = \frac{1}{n}\frac{\mathbf{G}^\top(\mathbf{X}\mathbf{X}^\top\circ\mathbf{I})\mathbf{G}}{\operatorname{tr}((\mathbf{X}\mathbf{X}^\top\circ\mathbf{I})\circ(\mathbf{G}\mathbf{G}^\top\circ\mathbf{I}))} \in \mathbb{R}^{m\times m}. \tag{19}$$

Under the Assumption 1, we have $(\mathbf{X}\mathbf{X}^\top)_{ii} = 1, i\in[d]$. Therefore, Eq. (18) and Eq. (19) can be simplified to

$$\boldsymbol{\Phi} = \mathbf{Z}_{\mathbf{X},\mathbf{G}}, \quad \boldsymbol{\Psi} = \frac{\mathbf{G}^\top\mathbf{G}}{n\eta},$$

where $\eta = \operatorname{tr}((\mathbf{X}\mathbf{X}^\top\circ\mathbf{I})\circ(\mathbf{G}\mathbf{G}^\top\circ\mathbf{I}))$ and $\mathbf{Z}_{\mathbf{X},\mathbf{G}} = \mathbf{X}^\top(\mathbf{G}\mathbf{G}^\top\circ\mathbf{I})\mathbf{X}$. Finally, the update rule of LNGD can be given by

$$\boldsymbol{\theta}^{k+1} = \boldsymbol{\theta}^k - \alpha[\eta(\mathbf{Z}_{\mathbf{X},\mathbf{G}}^k)^{-1}\otimes((\mathbf{G}^k)^\top\mathbf{G}^k)^{-1}](\mathbf{J}^k)^\top(\mathbf{v}^k - \mathbf{y}).$$

To analyze the global convergence of LNGD, we need the following two conditions as given in [34].

**Condition 1.** *The matrix $\mathbf{G}^0(\mathbf{G}^0)^\top$ is positive definite.*

To verify this condition, we need the following two lemmas.

**Lemma 1.** *Define $\mathbf{K}_{i,j}^\infty = \mathbb{E}_{\boldsymbol{\theta} \sim \mathcal{N}(\mathbf{0}, \nu^2 \mathbf{I})}[\mathbb{I}\{\boldsymbol{\theta}^\top \mathbf{x}_i \geq 0, \boldsymbol{\theta}^\top \mathbf{x}_j \geq 0\}], \ i, j \in [n]$, then we have the matrix $\mathbf{K}^\infty$ is strictly positive definite.*

The result of this lemma has been given and discussed in [34]. We define $\lambda_{\mathbf{K}} = \lambda_{\min}(\mathbf{K}^\infty) \geq 0$ and matrix $\mathbf{K}$ whose entry is given by

$$\mathbf{K}_{ij} = \frac{1}{m}\sum_{r=1}^m \mathbb{I}\{\boldsymbol{\theta}_r^\top \mathbf{x}_i \geq 0, \boldsymbol{\theta}_r^\top \mathbf{x}_j \geq 0\} = (\mathbf{G}\mathbf{G}^\top)_{ij}, \ \forall \, i, j \in [n].$$

Then we can show Condition 1 holds by the following lemma.

**Lemma 2.** *If $m = \Omega\left(\frac{n}{\lambda_{\mathbf{K}}}\log\frac{n}{\varepsilon}\right)$, we have with probability at least $1 - \varepsilon$ that $\lambda_{\min}(\mathbf{K}(0)) \geq \frac{3}{4}\lambda_{\mathbf{K}}$.*

*Proof.* Note that $\mathbf{K}(0)$ can be written as the sum of random symmetric matrices

$$\mathbf{K}(0) = \sum_{r=1}^m \mathbf{K}(\boldsymbol{\theta}_r), \quad \mathbf{K}_{ij}(\boldsymbol{\theta}_r) = \frac{1}{m}\mathbb{I}\{\theta_r^\top \mathbf{x}_i \geq 0, \theta_r^\top \mathbf{x}_j \geq 0\}.$$

Furthermore, we have $\mathbf{K}(\boldsymbol{\theta}_r)$ are positive semi-definite and $\|\mathbf{K}(\boldsymbol{\theta}_r)\|_2 \leq \text{tr}(\mathbf{K}(\boldsymbol{\theta}_r)) \leq \frac{n}{m}$. Thus, by the matrix Chernoff bound [37], we can obtain

$$\mathbb{P}\left[\lambda_{\min}(\mathbf{K}(0)) \leq (1 - \frac{1}{4})\lambda_{\mathbf{K}}\right] \leq n \exp\left(-\frac{1}{4^2}\frac{\lambda_{\mathbf{K}}m}{n}\right).$$

Let $\varepsilon = n\exp\left(-\frac{1}{4^2}\frac{\lambda_{\mathbf{K}}m}{n}\right)$, we have $m = \Omega\left(\frac{n}{\lambda_{\mathbf{K}}}\log\frac{n}{\varepsilon}\right)$. Proof complete. $\qquad\square$

This proof is similar to the Lemma 6 in [34], the difference is the definition of $\mathbf{K}$. For completeness of the proof, we also give the detailed proof here. By this lemma, we have $\lambda_{\min}(\mathbf{G}(0)\mathbf{G}(0)^\top) = \lambda_{\min}(\mathbf{K}(0)) \geq \frac{3}{4}\lambda_{\mathbf{K}} > 0$, which implies that Condition 1 holds. Next, we give the other condition.

**Condition 2.** *For all parameters $\boldsymbol{\theta}$ that satisfy $\|\boldsymbol{\theta} - \boldsymbol{\theta}^0\|_2 \leq \frac{2\|\mathbf{y} - \mathbf{v}^0\|_2}{\sqrt{\lambda_{\mathbf{K}}/2}}\kappa_{\mathbf{Z}_{\mathbf{X},\mathbf{G}}}$, there exists $0 \leq c < \frac{1}{2}$ such that*

$$\|\mathbf{J} - \mathbf{J}^0\|_2 \leq \frac{\sqrt{2}c}{4}\frac{\sqrt{\lambda_{\mathbf{K}}}}{\kappa_{\mathbf{Z}_{\mathbf{X},\mathbf{G}}}}.$$

To show this condition holds, we need the following lemma.

**Lemma 3.** *[34] For all weight vectors $\boldsymbol{\theta}$ that satisfy $\|\boldsymbol{\theta} - \boldsymbol{\theta}^0\|_2 \leq R$, we have probability at least $1 - \varepsilon$ that*

$$\|\mathbf{J} - \mathbf{J}^0\|_2^2 \leq \|\mathbf{J} - \mathbf{J}^0\|_{\mathbf{F}}^2 \leq \frac{2nR^{2/3}}{\nu^{2/3}\varepsilon^{2/3}m^{1/3}}.$$

By taking $R = \frac{2\|\mathbf{y} - \mathbf{v}^0\|_2}{\sqrt{\lambda_{\mathbf{K}}/2}}\kappa_{\mathbf{Z}_{\mathbf{X},\mathbf{G}}}$, we have $\|\mathbf{J} - \mathbf{J}^0\|_2^2 \leq \frac{64^{1/3}n\|\mathbf{y} - \mathbf{v}^0\|_2^{2/3}\kappa_{\mathbf{Z}_{\mathbf{X},\mathbf{G}}}^{2/3}}{\nu^{2/3}\varepsilon^{2/3}m^{1/3}\lambda_{\mathbf{K}}^{1/3}}$. Therefore, if we let $m = \Omega\left(\frac{n^3\|\mathbf{y} - \mathbf{v}^0\|_2^2\kappa_{\mathbf{Z}_{\mathbf{X},\mathbf{G}}}^8}{\nu^2\varepsilon^2\lambda_{\mathbf{K}}^4}\right)$, the Condition 2 holds. What's more, we have probability at least $1 - \varepsilon$ that $\|\mathbf{y} - \mathbf{v}^0\|_2^2 = \mathcal{O}\left(\frac{n}{\varepsilon}\right)$, which has been given in [31]. Thus we can write the condition of $m$ as $m = \Omega\left(\frac{n^4\kappa_{\mathbf{Z}_{\mathbf{X},\mathbf{G}}}^8}{\nu^2\varepsilon^3\lambda_{\mathbf{K}}^4}\right)$.

Before giving the main result, we first give some necessary lemmas.

**Lemma 4.** *If $m = \Omega\left(\frac{n^4\kappa_{\mathbf{Z}_{\mathbf{X},\mathbf{G}}}^8}{\nu^2\varepsilon^3\lambda_{\mathbf{K}}^4}\right)$, for all parameters $\boldsymbol{\theta}$ that satisfy $\|\boldsymbol{\theta} - \boldsymbol{\theta}^0\|_2 \leq \frac{2\|\mathbf{y} - \mathbf{v}^0\|_2}{\sqrt{\lambda_{\mathbf{G}}/2}}\kappa_{\mathbf{Z}_{\mathbf{X},\mathbf{G}}}$, we have probability at least $1 - \varepsilon$ that $\lambda_{\min}(\mathbf{K}) \geq \sqrt{\lambda_{\mathbf{K}}/2}$.*

*Proof.* Combine Eq. (15), Eq. (17) and Assumption 1, we have $\|\mathbf{G} - \mathbf{G}(0)\|_{\mathbf{F}}^2 \leq \|\mathbf{J} - \mathbf{J}^0\|_{\mathbf{F}}^2$. Let $m = \Omega\left(\frac{n^4 \kappa_{\mathbf{Z}_{\mathbf{X},\mathbf{G}}}^8}{\nu^2 \varepsilon^3 \lambda_{\mathbf{K}}^4}\right)$ and $R = \frac{2\|\mathbf{y} - \mathbf{v}^0\|_2}{\sqrt{\lambda_{\mathbf{K}}/2}} \kappa_{\mathbf{Z}_{\mathbf{X},\mathbf{G}}}$, by Condition 2 and Lemma 3 we have

$$\|\mathbf{G} - \mathbf{G}(0)\|_2^2 \leq \|\mathbf{G} - \mathbf{G}(0)\|_{\mathbf{F}}^2 \leq \|\mathbf{J} - \mathbf{J}^0\|_{\mathbf{F}}^2$$
$$\leq \frac{c}{8} \frac{\lambda_{\mathbf{K}}}{\kappa_{\mathbf{Z}_{\mathbf{X},\mathbf{G}}}^2} \leq \frac{c}{8} \lambda_{\mathbf{K}}.$$

Therefore,

$$\sigma_{\min}(\mathbf{G}) \geq \sigma_{\min}(\mathbf{G}(0)) - \|\mathbf{G} - \mathbf{G}(0)\|_2$$
$$\geq \sqrt{\frac{3}{4}}\sqrt{\lambda_{\mathbf{K}}} - \sqrt{\frac{c}{8}}\sqrt{\lambda_{\mathbf{K}}} \geq \frac{\sqrt{2}}{2}\sqrt{\lambda_{\mathbf{K}}},$$

Note that the large $m$ is, the smaller $c$ is. Therefore, we can choose a slight larger $m$ satisfying this inequality. So we have

$$\lambda_{\min}(\mathbf{K}) = \lambda_{\min}(\mathbf{G}\mathbf{G}^\top) \geq \sqrt{\frac{\lambda_{\mathbf{K}}}{2}}.$$

Proof complete. $\qquad\square$

**Lemma 5.** *[38] Let* $\mathbf{A}$ *and* $\mathbf{B}$ *be two positive define matrices, we have*

$$\lambda_{\max}(\mathbf{A} \circ \mathbf{B}) \leq \left(\max_i \mathbf{A}_{ii}\right) \lambda_{\max}(\mathbf{B}),$$
$$\lambda_{\min}(\mathbf{A} \circ \mathbf{B}) \geq \left(\min_i \mathbf{A}_{ii}\right) \lambda_{\min}(\mathbf{B}).$$

**Lemma 6.** *[39] Let* $\otimes$ *denote the Kronecker product and* $*$ *denote the column-wise Khatri-Rao product, we have*

$$(\mathbf{A} \otimes \mathbf{B})(\mathbf{C} * \mathbf{D}) = \mathbf{A}\mathbf{C} * \mathbf{B}\mathbf{D},$$
$$(\mathbf{A} * \mathbf{B})^\top (\mathbf{A} * \mathbf{B}) = \mathbf{A}^\top \mathbf{A} \circ \mathbf{B}^\top \mathbf{B}.$$

Now, we give the convergence analysis of LNGD.

**Theorem 3.** *(Convergence rate of LNGD) Under the Assumption 1 and the assumption that* $\text{rank}(\mathbf{X}) = d$. *If we set the number of hidden units* $m = \Omega\left(\frac{n^4 \kappa_{\mathbf{Z}_{\mathbf{X},\mathbf{G}}}^8}{\nu^2 \varepsilon^3 \lambda_{\mathbf{G}}^4}\right)$, *we i.i.d initialize* $\boldsymbol{\theta}_r \sim \mathcal{N}(0, \nu\mathbf{I})$, $a_r \sim \text{unif}[\{-1, +1\}]$ *for any* $r \in [m]$, *and we set the step size* $\alpha \leq \frac{(1-2c)}{(1+c)^2}$. *Then with probability at least* $1 - \varepsilon$ *over the random initialization, we have for* $k = 0, 1, 2, \ldots$

$$\|\mathbf{y} - \mathbf{v}^k\|_2^2 \leq (1 - \alpha)^k \|\mathbf{y} - \mathbf{v}^0\|_2^2.$$

*Proof.* Consider the predictive error at the $(k+1)$-th iteration, we have

$$\|\mathbf{y} - \mathbf{v}^{k+1}\|_2^2 = \|\mathbf{y} - \mathbf{v}^k + \mathbf{v}^k - \mathbf{v}^{k+1}\|_2^2$$
$$= \|\mathbf{y} - \mathbf{v}^k\|_2^2 - 2(\mathbf{y} - \mathbf{v}^k)^\top(\mathbf{v}^{k+1} - \mathbf{v}^k) + \|\mathbf{v}^{k+1} - \mathbf{v}^k\|_2^2. \qquad (20)$$

Next, we need to estimate the bound of prediction

$$\mathbf{v}^{k+1} - \mathbf{v}^k = \mathbf{v}(\boldsymbol{\theta}^k - \alpha(\mathbf{F}^k)^{-1}(\mathbf{J}^k)^\top(\mathbf{v}^k - \mathbf{y})) - \mathbf{v}^k$$

$$= -\int_{\xi=0}^1 \alpha \mathbf{J}^\xi(\mathbf{F}^k)^{-1}(\mathbf{J}^k)^\top(\mathbf{v}^k - \mathbf{y})\xi$$

$$= -\int_{\xi=0}^1 \alpha \mathbf{J}^k(\mathbf{F}^k)^{-1}(\mathbf{J}^k)^\top(\mathbf{v}^k - \mathbf{y})\xi$$

$$+ \int_{\xi=0}^1 \alpha(\mathbf{J}^k - \mathbf{J}^\xi)(\mathbf{F}^k)^{-1}(\mathbf{J}^k)^\top(\mathbf{v}^k - \mathbf{y})\xi$$

$$= \underbrace{-\alpha \mathbf{J}^k(\mathbf{F}^k)^{-1}(\mathbf{J}^k)^\top(\mathbf{v}^k - \mathbf{y})}_{\text{Term 1}}$$

$$+ \underbrace{\alpha\left(\int_{\xi=0}^1 (\mathbf{J}^k - \mathbf{J}^\xi)\xi\right)(\mathbf{F}^k)^{-1}(\mathbf{J}^k)^\top(\mathbf{v}^k - \mathbf{y})}_{\text{Term 2}},$$

where $\mathbf{J}^\xi = \frac{\partial \mathbf{v}(\boldsymbol{\theta}^\xi)}{\partial \boldsymbol{\theta}^\xi}$, and $\boldsymbol{\theta}^\xi = \xi\boldsymbol{\theta}^k + (1-\xi)\boldsymbol{\theta}^{k+1} = \boldsymbol{\theta}^k - \xi\alpha(\mathbf{F}^k)^{-1}(\mathbf{J}^k)^\top(\mathbf{v}^k - \mathbf{y})$.

We first analyse Term 1. We omit the index $k$ in $\mathbf{J}$, $\mathbf{G}$ and $\mathbf{F}$ for simplicity.

$$\text{Term 1} = -\alpha\mathbf{J}\mathbf{F}^{-1}\mathbf{J}^\top(\mathbf{v}^k - \mathbf{y})$$
$$=\alpha(\mathbf{X} \star \mathbf{G})[\eta\mathbf{Z}_{\mathbf{X},\mathbf{G}}^{-1} \otimes (\mathbf{G}^\top\mathbf{G})^{-1}](\mathbf{X}^\top \ast \mathbf{G}^\top)(\mathbf{y} - \mathbf{v}^k)$$
$$=\alpha(\eta\mathbf{X}\mathbf{Z}_{\mathbf{X},\mathbf{G}}^{-1}\mathbf{X}^\top \circ \mathbf{G}(\mathbf{G}^\top\mathbf{G})^{-1}\mathbf{G}^\top)(\mathbf{y} - \mathbf{v}^k)$$
$$=\alpha(\eta\mathbf{X}\mathbf{Z}_{\mathbf{X},\mathbf{G}}^{-1}\mathbf{X}^\top \circ \mathbf{I})(\mathbf{y} - \mathbf{v}^k),$$

The second equation follows the update rule of LNGD. The third equation is obtained according to the properties of Kronecker, Hadamard and Khatri-Rao products given in Lemma 6. The last equation uses the definition of generalized inverse as given by Eq. (16). By Lemma 5, we have

$$\lambda_{\max}(\mathbf{X}\mathbf{Z}_{\mathbf{X},\mathbf{G}}^{-1}\mathbf{X}^\top \circ \mathbf{I}) \le \max_i(\mathbf{X}\mathbf{Z}_{\mathbf{X},\mathbf{G}}^{-1}\mathbf{X}^\top)_{ii}\lambda_{\max}(\mathbf{I})$$
$$\le \lambda_{\max}(\mathbf{Z}_{\mathbf{X},\mathbf{G}}^{-1}) \max_i(\mathbf{X}\mathbf{X}^\top)_{ii} = \frac{1}{\lambda_{\min}(\mathbf{Z}_{\mathbf{X},\mathbf{G}})}. \tag{21}$$

Therefore, we can bound Term 1 by

$$\|\text{Term}_1\|_2 = \|\alpha(\eta\mathbf{X}\mathbf{Z}_{\mathbf{X},\mathbf{G}}^{-1}\mathbf{X}^\top \circ \mathbf{I})(\mathbf{y} - \mathbf{v}^k)\|_2$$
$$\le \alpha\eta\|\mathbf{X}\mathbf{Z}_{\mathbf{X},\mathbf{G}}^{-1}\mathbf{X}^\top \circ \mathbf{I}\|_2\|\mathbf{y} - \mathbf{v}^k\|_2 \le \frac{\alpha\eta}{\lambda_{\min}(\mathbf{Z}_{\mathbf{X},\mathbf{G}})}\|\mathbf{y} - \mathbf{v}^k\|_2. \tag{22}$$

Based on the Condition 2, we have the following inequality

$$\left\|\int_{\xi=0}^1 (\mathbf{J}^k - \mathbf{J}^\xi)\xi\right\|_2 \le \int_{\xi=0}^1 \|\mathbf{J}^k - \mathbf{J}^\xi\|_2\xi \le \|\mathbf{J}^{k+1} - \mathbf{J}^k\|_2$$
$$\le\|\mathbf{J}^{k+1} - \mathbf{J}^0\|_2 + \|\mathbf{J}^k - \mathbf{J}^0\|_2 \tag{23}$$
$$\le\frac{\sqrt{2}c}{2}\frac{\sqrt{\lambda_{\mathbf{K}}}}{\kappa_{\mathbf{Z}_{\mathbf{X},\mathbf{G}}}} \le \frac{c}{\kappa_{\mathbf{Z}_{\mathbf{X},\mathbf{G}}}}\sqrt{\lambda_{\min}(\mathbf{G}\mathbf{G}^\top)}.$$

Next, we bound Term 2. By Eq. (23), we have

$$\|\text{Term 2}\|_2 = \left\|\alpha\left(\int_{\xi=0}^1 (\mathbf{J}^k - \mathbf{J}^\xi)\xi\right)(\mathbf{F}^k)^{-1}(\mathbf{J}^k)^\top(\mathbf{v}^k - \mathbf{y})\right\|_2$$
$$\le\frac{c\alpha}{\kappa_{\mathbf{Z}_{\mathbf{X},\mathbf{G}}}}\sqrt{\lambda_{\min}(\mathbf{G}\mathbf{G}^\top)}\|(\mathbf{F}^k)^{-1}(\mathbf{J}^k)^\top\|_2\|\mathbf{y} - \mathbf{v}^k\|_2$$
$$=\frac{c\alpha}{\kappa_{\mathbf{Z}_{\mathbf{X},\mathbf{G}}}}\sqrt{\lambda_{\min}(\mathbf{G}\mathbf{G}^\top)}\|(\eta\mathbf{Z}_{\mathbf{X},\mathbf{G}}^{-1} \otimes (\mathbf{G}^\top\mathbf{G})^{-1})(\mathbf{X}^\top \ast \mathbf{G}^\top)\|_2\|\mathbf{y} - \mathbf{v}^k\|_2$$
$$=\frac{c\alpha}{\kappa_{\mathbf{Z}_{\mathbf{X},\mathbf{G}}}}\sqrt{\lambda_{\min}(\mathbf{G}\mathbf{G}^\top)}\|\eta\mathbf{Z}_{\mathbf{X},\mathbf{G}}^{-1}\mathbf{X}^\top \ast (\mathbf{G}^\top\mathbf{G})^{-1}\mathbf{G}^\top\|_2\|\mathbf{y} - \mathbf{v}^k\|_2$$
$$=\frac{c\alpha}{\kappa_{\mathbf{Z}_{\mathbf{X},\mathbf{G}}}}\sqrt{\lambda_{\min}(\mathbf{G}\mathbf{G}^\top)}\|\eta\mathbf{Z}_{\mathbf{X},\mathbf{G}}^{-1}\mathbf{X}^\top \ast \mathbf{G}^\top(\mathbf{G}\mathbf{G}^\top)^{-1}(\mathbf{G}\mathbf{G}^\top)^{-1}\mathbf{G}\mathbf{G}^\top\|_2\|\mathbf{y} - \mathbf{v}^k\|_2$$
$$=\frac{c\alpha\eta}{\kappa_{\mathbf{Z}_{\mathbf{X},\mathbf{G}}}}\sqrt{\lambda_{\min}(\mathbf{G}\mathbf{G}^\top)}\|\mathbf{Z}_{\mathbf{X},\mathbf{G}}^{-1}\mathbf{X}^\top \ast \mathbf{G}^\top(\mathbf{G}\mathbf{G}^\top)^{-1}\|_2\|\mathbf{y} - \mathbf{v}^k\|_2. \tag{24}$$

Define $\Delta = \mathbf{Z}_{\mathbf{X},\mathbf{G}}^{-1}\mathbf{X}^\top \ast \mathbf{G}^\top(\mathbf{G}\mathbf{G}^\top)^{-1}$ , then we have

$$\|\Delta\|_2 = \sigma_{\max}(\Delta) = \sqrt{\lambda_{\max}(\Delta^\top\Delta)}$$
$$= \sqrt{\lambda_{\max}(\mathbf{X}\mathbf{Z}_{\mathbf{X},\mathbf{G}}^{-1}\mathbf{Z}_{\mathbf{X},\mathbf{G}}^{-1}\mathbf{X}^\top \circ (\mathbf{G}\mathbf{G}^\top)^{-1})}. \tag{25}$$

Similar to Eq. (21), by Lemma (5) we can prove

$$\|\Delta\|_2 \le \frac{1}{\lambda_{\min}(\mathbf{Z}_{\mathbf{X},\mathbf{G}})}\frac{1}{\sqrt{\lambda_{\min}(\mathbf{G}\mathbf{G}^\top)}}. \tag{26}$$

Combine Eq. (24) and Eq. (26), we have

$$\|\text{Term}_2\|_2 \leq \frac{c\alpha\eta}{\kappa_{\mathbf{Z}_{\mathbf{X},\mathbf{G}}}} \sqrt{\lambda_{\min}(\mathbf{G}\mathbf{G}^\top)} \frac{1}{\lambda_{\min}(\mathbf{Z}_{\mathbf{X},\mathbf{G}})} \frac{1}{\sqrt{\lambda_{\min}(\mathbf{G}\mathbf{G}^\top)}} \|\mathbf{y} - \mathbf{v}^k\|_2$$

$$= \frac{c\alpha\eta}{\lambda_{\max}(\mathbf{Z}_{\mathbf{X},\mathbf{G}})} \|\mathbf{y} - \mathbf{v}^k\|_2.$$

(27)

Combine Eq. (20), Eq. (27) and Eq. (22), we can obtain

$$\|\mathbf{y} - \mathbf{v}^{k+1}\|_2^2 = \|\mathbf{y} - \mathbf{v}^k\|_2^2 - 2(\mathbf{y} - \mathbf{v}^k)^\top(\mathbf{v}^{k+1} - \mathbf{v}^k) + \|\mathbf{v}^{k+1} - \mathbf{v}^k\|_2^2$$

$$= \|\mathbf{y} - \mathbf{v}^k\|_2^2 - 2\alpha(\mathbf{y} - \mathbf{v}^k)^\top \mathbf{J}^k(\mathbf{F}^k)^{-1}(\mathbf{J}^k)^\top(\mathbf{y} - \mathbf{v}^k)$$

$$+ 2\alpha(\mathbf{y} - \mathbf{v}^k)^\top \left(\int_{\xi=0}^1 (\mathbf{J}^k - \mathbf{J}^\xi)\xi\right)(\mathbf{F}^k)^{-1}(\mathbf{J}^k)^\top(\mathbf{y} - \mathbf{v}^k)$$

$$+ \|\mathbf{v}^{k+1} - \mathbf{v}^k\|_2^2$$

$$\leq \left(1 - \frac{2\alpha\eta}{\lambda_{\max}(\mathbf{Z}_{\mathbf{X},\mathbf{G}})} + \frac{2c\alpha\eta}{\lambda_{\max}(\mathbf{Z}_{\mathbf{X},\mathbf{G}})} + \frac{\alpha^2\eta^2}{\lambda_{\min}^2(\mathbf{Z}_{\mathbf{X},\mathbf{G}})}\right.$$

$$\left. + \frac{2c\alpha^2\eta^2}{\lambda_{\max}(\mathbf{Z}_{\mathbf{X},\mathbf{G}})\lambda_{\min}(\mathbf{Z}_{\mathbf{X},\mathbf{G}})} + \frac{c^2\alpha^2\eta^2}{\lambda_{\max}^2(\mathbf{Z}_{\mathbf{X},\mathbf{G}})}\right)\|\mathbf{y} - \mathbf{v}^k\|_2^2.$$

In the last second inequality, we use the fact that $\lambda_{\min}(\mathbf{X}\mathbf{Z}_{\mathbf{X},\mathbf{G}}^{-1}\mathbf{X}^\top \circ \mathbf{I}) \geq \frac{1}{\lambda_{\max}(\mathbf{Z}_{\mathbf{X},\mathbf{G}})}$. Let

$$-\frac{\alpha\eta}{\lambda_{\max}(\mathbf{Z}_{\mathbf{X},\mathbf{G}})} + \frac{2c\alpha\eta}{\lambda_{\max}(\mathbf{Z}_{\mathbf{X},\mathbf{G}})} + \frac{\alpha^2\eta^2}{\lambda_{\min}^2(\mathbf{Z}_{\mathbf{X},\mathbf{G}})}$$

$$+ \frac{2c\alpha^2\eta^2}{\lambda_{\max}(\mathbf{Z}_{\mathbf{X},\mathbf{G}})\lambda_{\min}(\mathbf{Z}_{\mathbf{X},\mathbf{G}})} + \frac{c^2\alpha^2\eta^2}{\lambda_{\max}^2(\mathbf{Z}_{\mathbf{X},\mathbf{G}})} \leq 0,$$

we have

$$\alpha \leq \frac{(1 - 2c)\lambda_{\max}(\mathbf{Z}_{\mathbf{X},\mathbf{G}})}{(1 + c)^2\eta} \leq \frac{(1 - 2c)}{(1 + c)^2},$$

and

$$\|\mathbf{y} - \mathbf{v}^k\|_2^2 \leq (1 - \alpha)^k\|\mathbf{y} - \mathbf{v}^0\|_2^2.$$

This completes the proof. □

So far, we have already proved Theorem 3 under the an assumption that the parameters stay close to the initialization point. We now verify this assumption by the following lemma.

**Lemma 7.** *If Conditions 1 and 2 hold, then as long as* $\lambda_{\min}(\mathbf{G}\mathbf{G}^\top) \geq \frac{1}{2}\lambda_{\mathbf{K}}$, *we have*

$$\|\boldsymbol{\theta}^{k+1} - \boldsymbol{\theta}^0\|_2 \leq \frac{2\|\mathbf{y} - \mathbf{v}^0\|_2}{\sqrt{\lambda_{\mathbf{K}}/2}}\kappa_{\mathbf{Z}_{\mathbf{X},\mathbf{G}}}.$$

*Proof.* By the update rule of LNGD, we have

$$\|\boldsymbol{\theta}^{k+1} - \boldsymbol{\theta}^0\|_2 = \left\|\sum_{t=0}^k \alpha(\mathbf{F}^t)^{-1}(\mathbf{J}^t)^\top(\mathbf{y} - \mathbf{v}^t)\right\|_2$$

$$\leq \alpha\sum_{t=0}^k \|\|\mathbf{Z}_{\mathbf{X},\mathbf{G}}^{-1}\mathbf{X}^\top * (\mathbf{G}^t)^\top(\mathbf{G}^t(\mathbf{G}^t)^\top)^{-1}\|_2\|\mathbf{y} - \mathbf{v}^t\|_2$$

$$\leq \alpha\sum_{t=0}^k \frac{1}{\lambda_{\min}(\mathbf{Z}_{\mathbf{X},\mathbf{G}})} \frac{1}{\sqrt{\lambda_{\mathbf{K}}/2}}\|\mathbf{y} - \mathbf{v}^t\|_2$$

$$\leq \alpha\sum_{t=0}^k \frac{\sqrt{2/\lambda_{\mathbf{K}}}}{\lambda_{\min}(\mathbf{Z}_{\mathbf{X},\mathbf{G}})}\left(1 - \frac{\alpha}{\lambda_{\max}(\mathbf{Z}_{\mathbf{X},\mathbf{G}})}\right)^{t/2}\|\mathbf{y} - \mathbf{v}^0\|_2$$

$$\leq \frac{2\|\mathbf{y} - \mathbf{v}^0\|_2}{\sqrt{\lambda_{\mathbf{K}}/2}} \frac{\lambda_{\max}(\mathbf{Z}_{\mathbf{X},\mathbf{G}})}{\lambda_{\min}(\mathbf{Z}_{\mathbf{X},\mathbf{G}})} = \frac{2\|\mathbf{y} - \mathbf{v}^0\|_2}{\sqrt{\lambda_{\mathbf{K}}/2}}\kappa_{\mathbf{Z}_{\mathbf{X},\mathbf{G}}}.$$

This completes the proof. □

# E Experiments

## E.1 Setup of CIFAR-10

The training of ResNet-18 [44] on the CIFAR-10 [40] dataset serves as a fundamental experiment in the field of image classification. In this subsection, we present a comparison of LNGD with several established baselines including SGD with momentum (referred to as SGD), ADAM [6], and KFAC. We follow the standard experimental settings and employ a commonly used data augmentation scheme involving random crop and horizontal flip. The initial learning rate is multiplied by 0.1 every 40 epochs. The update intervals for the curvature matrix and inverse matrix correlating with KFAC and LNGD are set to be 100. All experimental runs are conducted over a duration of 200 epochs.

## E.2 Setup of ImageNet

The implementation of ResNet50 [44] follows the TensorFlow version which can be found in the website [3]. We use the linear warmup strategy [41] in the first 5 epochs for SGD, ADAM and KFAC. The update intervals for the curvature matrix and inverse matrix correlating with KFAC and LNGD are set to be 500. For SGD and Adam, the max epoch is set to be 80, while for KFAC and LNGD, the max epoch is set to be 50. SGD uses the cosine learning rate updating strategy and is set to be $\alpha_t = 0.001 + 0.5 * (\alpha_0 - 0.001) * (1 + \cos(2 * 0.47 * \pi * t/max\_epoch))$, where $t$ is the number of epochs. For Adam, KFAC and LNGD, the learning rate uses the exponential updating strategy $\alpha_t = \alpha_0 * (1 - t/max\_epoch)^E$, where $E$ is decay rate $\in \{2, 3, 4, 5, 6\}$. $\alpha_0$ is the initial learning rate tuned using a grid search with values $\alpha \in \{1e-4, 1e-3, \ldots, 1\}$.

## E.3 Ablation Analysis

### E.3.1 Setup

In this subsection, to further elucidate the contributions of distinct components within the LNGD, a series of ablation studies are performed. The ablation experiments aim to isolate the effects of adaptive learning rate and sampling optimization on the LNGD's performance. The variant denoted as LNGD-lr corresponds to the iteration of the algorithm that employs an adaptive learning rate, but does not incorporate sampling optimization. Conversely, LNGD-sample represents the iteration that utilizes sampling optimization, but does not implement an adaptive learning rate. These ablation studies are executed on the ImageNet-1K dataset, All hyperparameters are maintained consistent with those outlined in the ImageNet training section.

### E.3.2 Results

The results of the ablation experiments, as shown in Fig.7 and Table5, reveal some interesting findings. Specifically, analyzing the training loss and testing accuracy versus epoch, we observe that LNGD-lr achieves the fastest decrease in training loss and the most rapid initial increase in testing accuracy within the initial few epochs. This can be attributed to the fact that LNGD-lr computes the exact Fisher information matrix at each epoch without using any approximation sampling strategy. However, this advantage comes at the cost of increased computational complexity, leading to a 15% increase in the time required to reach a top-1 testing accuracy of 75.9% compared to LNGD, which employs both the sampling approximation strategy and the adaptive learning rate strategy at each layer. Moreover, LNGD-lr also takes 5% more time compared to LNGD-sample, which only utilizes the sampling approximation strategy. Notably, LNGD-sample exhibits the slowest decrease in training loss and increase in testing accuracy during the initial epochs due to its approximation sampling of the Fisher information matrix at each step. Nevertheless, when considering the time dimension, LNGD-sample still achieves a faster speed compared with LNGD-lr in reaching a final testing accuracy of 75.9% due to the significant reduction in the computation of the exact Fisher information matrix. In contrast to LNGD, LNGD-sample takes 9% more time to reach testing accuracy of 75.9% due to the absence of automatic scaling learning rate. In conclusion, considering the constraints of limited computational resources and time, LNGD demonstrates superior optimizing performance.

---

[3]https://github.com/google-deepmind/dm-haiku/tree/main/examples/imagenet

Table 5: Detailed statistics of abalation study when top-1 testing accuracy achieves 75.9%.

|  | Epoch | Total Time | Time Per Epoch | Acceleration | Best Test Acc |
|---|---|---|---|---|---|
| LNGD-lr | 35 | 7.43h | 764.39s | 13% | 76.50% |
| LNGD-sample | 41 | 7.06h | 619.86s | 9% | 76.57% |
| LNGD | 36 | 6.46h | 646.44s |  | 76.73% |

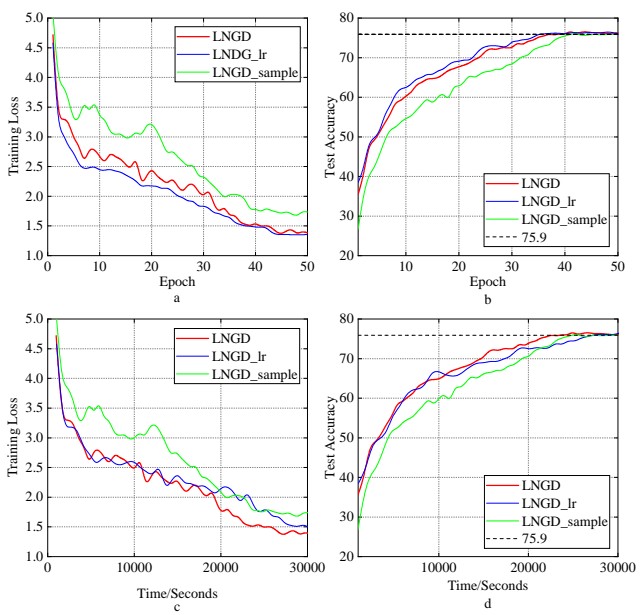

Figure 7: The optimization performance of variants of LNGD.

## E.4 Results of More Comparisons

Table 6: Detailed statistics on CIFAR-10 when top-1 testing accuracy achieves 91%.

|  | Epoch | Total Time | Time Per Epoch | Acceleration |
|---|---|---|---|---|
| SGD | 79 | 268.67s | 3.4s | 29% |
| ADAM | 72 | 248.83s | 3.77s | 23% |
| KFAC | 45 | 241.86s | 5.87s | 21% |
| EKFAC | 41 | 247.64s | 6.04s | 23% |
| TKFAC | 39 | 239.20s | 5.98s | 20% |
| NG+ | 40 | 204.45s | 5.11s | 7% |
| LNGD | 36 | 189.69s | 5.08s |  |

In order to further validate the effectiveness of LNGD, we conduct additional experiments on the CIFAR-10 dataset, in which three methods including EKFAC [22], TKFAC [15], and NG+ [17] are added for comparison. The detailed statistics are presented in Table 6. From this table, we observe that LNGD achieves a testing accuracy of 91% with the fewest epochs and the shortest total time. Furthermore, LNGD exhibits the smallest computational time per epoch. Additionally, due to the efficient Fisher information matrix approximation strategy adopted by NG+, it can significantly reduce the computational time compared to EKFAC and TKFAC.

