# OpenReview forum: "A Layer-Wise Natural Gradient Optimizer for Training Deep Neural Networks"
_NeurIPS.cc/2024/Conference — NeurIPS 2024 poster_

### Official Review · Reviewer_Mvn4 · 2024-07-07

**Soundness:** 3
**Presentation:** 3
**Contribution:** 3
**Rating:** 7
**Confidence:** 3

**Summary:**

The proposed approach uses the block diagonal form of the Fischer Information Matrix (FIM) which is achieved by computing this FIM with every layer in the network sequentially. In addition, the space complexity is limited by consider the diagonal form of this FIM matrix for each layer and its respective gradient outer products.

**Strengths:**

1. The authors find a clever way of approximating the Fischer information matrix by only consider the block diagonal form of the FIM. In addition, due to i.i.d. assumptions, the block diagonal reduces to a diagonal matrix for
$\Psi_l \otimes \Phi_l$
2. The adaptive learning rate for each layer further makes it an interesting way of optimizing the weights.

**Weaknesses:**

The reviewer has actually found it difficult to find any weaknesses with the paper. It is really well written, with sufficient proofs and sufficient results to support their claims.

**Questions:**

1. What is $\mathbf{s}_l$ in line 136 ?
2. This reveiwer would like to know what's the difference between the proposed approach and https://ojs.aaai.org/index.php/AAAI/article/view/16867 ?

**Limitations:**

This is not really a limitation - Given that this paper improves upon fundamental results significantly, an open source implementation of this approach would a be really good to have.

---

> ### Author Rebuttal · Authors · 2024-08-07
>
> Q1: What is $s_l$ in line 136 ?
>
> A1: Sorry for the unclear description, $s_l= W_l a_{l-1}$. We will correct it in the final version.
>
> Q2: This reveiwer would like to know what's the difference between the proposed approach and https://ojs.aaai.org/index.php/AAAI/article/view/16867?
>
> A2: Thanks for your question. The THOR, which is proposed in https://ojs.aaai.org/index.php/AAAI/article/view/16867, mainly considers reducing the computational cost of natural gradient descent through two aspects. On the one hand, THOR gradually increasess the updating frequency of the inverse matrix of Fisher information matrix (FIM) and proposes a trace-based updating rule for FIM for each layer. On the other hand, THOR approximates the approximated FIM obtained by KFAC as some smaller matrices by splitting matrix dimensions. Our proposed LNGD first gives a layer-wise sample method to more efficiently compute each block matrix corresponding to each layer and proposes a novel approximate scheme of the FIM. Furthermore, LNGD also adopts an adaptive layer-wise learning rate to speed up training. The contributions and ideas of our proposed LNGD are different from THOR.

---

> ### Comment · Reviewer_Mvn4 · 2024-08-09
> **Thank you for feedback**
>
> This reviewer thanks the authors for their feedback.
>
> This reviewer upholds his/her decision on the paper. There is solid contribution
>
> However, here are some minor changes the reviewer would suggest
> 1. Make the clear distinction between THOR and the proposed approach. At a first glance, the methods seem to have a lot of overlap, albeit the contributions might be different. It would be interesting to see a comparative performance of THOR against LNGD as well (if the code is reproducible and available to use).
> 2. Reiterating on reproducibility - It would be interesting to release a reproducible version of this code to the community.
> 3. Absolute difference between the exact FIM and the proposed approach FIM $\textbf{along the diagonal}$ - It would be interesting to see how different the diagonal elements of the FIM are from the exact version. It would be great if that information is provided in the final version of the paper as well.

---

> > ### Author Response · Authors · 2024-08-12
> >
> > Thank you for your new valuable comments and suggestions.
> >
> > We will add explanations in the final version to make a clear distinction between THOR and LNGD. We will also try to perform experiments to compare their performance.
> >
> > As for the code, we are currently undergoing the approval process for the open-source workflow in compliance with the company's regulations, and it will be open-sourced shortly.
> >
> > We will add figures and explanations in the final version to illustrate the difference between the exact FIM and the approximated FIM in LNGD along the diagonal.

---

### Official Review · Reviewer_gNTY · 2024-07-10

**Soundness:** 4
**Presentation:** 4
**Contribution:** 2
**Rating:** 4
**Confidence:** 3

**Summary:**

The authors propose a novel way to approximate the Fisher information matrix. They do this by starting with the block diagonal approximation of the Fisher information matrix which they compute using a novel layer-wise sampling method without performing a complete backpropagation. Then they approximate each block as a Kronecker product of two smaller matrices, where one is diagonal. They also keep the traces equal before and after the approximation. They call this new method the Layer-wise Natural Gradient Descent (LNGD). They further propose a new adaptive layer-wise learning rate that accelerates training. The authors also provide global convergence analysis of LNGD. At the end, they show extensive experiments on image classification and machine translation tasks, demonstrating their method is competitive.

**Strengths:**

- The idea of factoring the Fisher information matrix as in Section 3.1 seems novel and interesting.

- The authors provide a convergence result in Theorem 2.

- The authors have a great experimental section on complicated datasets. Their method outperforms all compared methods in both epoch and wallclock time.

**Weaknesses:**

- The biggest and most glaring weakness is that the paper doesn't compare with recent algorithms, some of which they even list in their own related works sections. The most recent algorithm that was compared is KFAC, which was published in 2015 -- almost a decade ago. If the authors provide comparisons to more recent algorithms (EKFAC, TKFAC, relevant extensions to convolutional neural networks, and NLP-related architectures, etc.), that might push it over to acceptance.

- There is the main idea (Section 3.1) of factoring the Fisher information matrix which seems novel. But then there are various techniques (Section 3.2 and 3.3) that are employed to make the method more effective. But it's hard to disentangle whether the main contribution to the numerical performance is due to the novel factoring that allows using the natural gradient, or the techniques such as adaptive learning rates based on the Hessian. As asked in the Questions section of this review, can you apply the adaptive learning rate techniques to KFAC? How does it compare?

**Questions:**

- I like Figure 1. But it would be interesting to also compare this with KFAC.

- In Section 3.2, doesn't having adaptive learning rates contradict the spirit of the natural gradient?
    - In the equation between line 172-173, isn't this computing the Taylor expansion, and thus also the Hessian? But I thought with the natural gradient and tricks to compute it, there should be no need to compute the Hessian?

- Can you also apply your adaptive learning rate strategy to KFAC?

**Limitations:**

The authors provide limitations for each theory.

---

> ### Author Rebuttal · Authors · 2024-08-07
>
> Q1:  The first part of weaknesses.
>
> A1: We are sincerely thankful for the valuable suggestions. Our primary objective is to propose and optimizer and achieve an optimal balance between the speed and accuracy of model training. Expanding upon established methodologies in the field of NGD, we propose a layer-wise sampling approach for efficiently computing the block matrix corresponding to each layer, as well as an adaptive layer-wise learning rate to improve training efficiency. In Appendix B of our manuscript, we provide comparisons of LNGD and KFAC as well as its recent variants (EKFAC and TKFAC). Additionally, in order to further substantiate the effectiveness of LNGD, we conducted additional experiments on CIFAR-10 and ImageNet using recent NGD methods, consistently demonstrating LNGD's superior performance in achieving comparable accuracy levels. Detailed experimental findings are available in our response to all reviewers, and we intend to include these pertinent results in the final version of our paper.
>
> Q2: The second part of weaknesses.
>
> A2: We appreciate the insightful suggestion. In order to provide a deeper understanding of the individual contributions of different components within the LNGD, a set of ablation studies have been conducted and are detailed in Appendix F.3 of our manuscript. These experiments are aimed at isolating the impact of adaptive learning rate and sampling optimization on the performance of the LNGD. Due to limitations in the rebuttal space, we will not elaborate further on this matter here and appreciate your understanding.
>
> Q3: I like Figure 1. But it would be interesting to also compare this with KFAC.
>
> A3: We have added the visualization results of KFAC in Figure 2, in which we only show partially enlarged figures due to page limitation.  From Figure (b) and (e), we can see that KFAC and LNGD can all emphasize the importance of the diagonal elements in the exact Fisher information matrix (FIM). In addition, it can also be seen clearly that KFAC still retains some elements near the main diagonal, while LNGD does not, which also reflects that LNGD provides an efficient approximation of FIM with less computational cost in comparison with KFAC.
>
> Q4: In Section 3.2, doesn't having adaptive learning rates contradict the spirit of the natural gradient?
>
> A4: Apologies for the lack of clarity in our presentation. We extend the current natural gradient methods by introducing a layer-wise sampling technique and an adaptive layer-wise learning rate to speed up the training process. In Section 3.2, we first discuss the necessity of an adaptive layer-wise learning rate, and then derive the formula for calculating it using a second-order Taylor expansion approach (i.e., Eq. (11)). Adhering to the NG principle, based on Eq. (11), we necessitate the computation of the Hessian matrix. However, across various loss functions such as cross-entropy or mean squared loss, the Fisher information matrix and the Hessian matrix are, in fact, equivalent. Hence, we proceed to compute the Fisher information matrix and utilize Eq. (12) to obtain the adaptive learning rate in LNGD.
>
> Q5: Can you also apply your adaptive learning rate strategy to KFAC?
>
> A5: Thanks for the valuable question. Our proposed adaptive learning rate strategy can also be applied to KFAC. As given in Appendix B, the main difference between KFAC and LNGD is the approximations of the block matrix $F_l$. KFAC approximates  $F_l$ as the Kronecker product of $A=E[a_{l-1}a_{l-1}^\top]$ and $ B=E[g_lg_l^\top]$. Our proposed LNGD
> approximates  $F_l$ as the Kronecker product of a matrix ${\bf\Phi}_l$ and a diagonal matrix ${\bf\Psi}_l$, which is computed by sampling from each layer, details are given in Section 3. The adaptive layer-wise learning rate is computed by Eq. (12), which would be the same for KFAC and LNGD. Hence, it is feasible to integrate the adaptive layer-wise learning rate into KFAC with minimal costs.

---

> > ### Comment · Reviewer_gNTY · 2024-08-09
> >
> > Thank you for your clarifications. I had a few more questions.
> >
> > Reply to Q.1 reply: For the first part of the weakness, yes I already saw Appendix B, but I was looking for performance comparisons. I see now that the authors have done this in the reply to all reviewers, thank you very much, I really appreciate it. But now can the authors expand on the experimental procedure with the new methods? For example, did the authors take the code from public code repositories and run it on the datasets? What hyperparameters were used? Also, apologies if I'm ignorant, but I just realized the oddity of using 91% and 75.9% as accuracy benchmarks. Why were these specific numbers chosen?
> >
> > Reply to Q.2 reply: For the second part of the weakness in my original review, I did see Section F.3 (which wasn't referenced in the main text, but perhaps should be referenced around line 260), but perhaps I wasn't being clear, my apologies. I was hoping to see the adaptive learning rates and sampling technique also applied to KFAC if applicable, and in light of the new experiments but me being sympathetic to the large amount of work involved, perhaps just to NG+. This is so we can examine if it's the novel factoring that shows impressive performance (which seems to me the main contribution of the paper), or the adaptive learning rate/sampling, or the new damping technique. I even see throughout the paper (Line 175-178 with adaptive learning rates, and Line 225-228 with damping) that adaptive learning rates and damping techniques are required to achieve optimal performance, so I would have hoped these advantages would be conferred onto KFAC as well, and now NG+ if they were also conferred to the proposed method, LNGD. I do realize this is a lot of work in a limited time, but I also believe this would make for a fairer comparison.
> >
> > Reply to Q.3 reply: From my eyes, and perhaps I'm looking at it incorrectly, but it seems KFAC is more accurate than LNGD? But are the authors arguing that LNGD achieves a "close-enough" approximation much faster?
> >
> > Reply to Q.4 reply: Thank you for the answer, it was enlightening. Apologies, but just to reclarify for my own ignorance: so to compute the adaptive learning rate, the Hessian matrix must be computed?
> >
> > Reply to Q.5 reply: See my answers to "Reply to Q.1" and "Reply to Q.2" above.

---

> > > ### Author Response · Authors · 2024-08-12
> > >
> > > Thank you for your new valuable comments and questions.
> > >
> > > Q 1. The new methods and experiments.
> > >
> > > A 1. Thank you for your question. We are sorry for the lack of detailed descriptions of the new methods’ setting in the reply. The code of EKFAC and NG+ are obtained from GitHub (links will be added in the final version since they are not permitted in our comment), and the code of TKFAC is obtained from its authors. The hyperparameters, such as the learning rate, the update of the curvature matrix strategy, the damping parameter, etc., are used according to the experimental settings in their papers and codes. We will add the detailed descriptions in the final version. Regarding the rationale for selecting accuracy benchmarks of 91% and 75.9%, we predominantly adhere to the settings established in the literature on natural gradient descent, specifically referencing several variants of KFAC.
> > >
> > > Q 2. KFAC with the adaptive learning rates and sampling technique.
> > >
> > > A 2. Thank you for your insightful question. We would like to take this opportunity to clarify the main contributions of our work. In this paper, we introduce a novel optimizer, LNGD, which builds upon existing natural gradient methodologies. Central to our approach are the novel layer-wise sampling strategy and the adaptive layer-wise learning rate strategy, which are essential components for executing the natural gradient optimization process within LNGD. As mentioned in our previous response, layer-wise sampling and adaptive layer-wise learning rates can also be integrated with KFAC and its variants, including EKFAC and TKFAC. However, once these techniques are incorporated into KFAC, EKFAC, and TKFAC, the resulting methods may be regarded as new variants of these methods rather than the original ones since some new techniques are incorporated.
> > >
> > > Certainly. To illustrate the effects of incorporating sampling strategy and adaptive learning rate into KFAC, we conducted experiments on the CIFAR-10 dataset. We denote L-KFAC as a model variant where the sampling strategy, the adaptive learning rate and the new damping have been integrated into KFAC. On the CIFAR-10 dataset, when the top-1 testing accuracy reached 91%, both L-KFAC and LNGD utilized 36 epochs; however, L-KFAC incurred a time cost of 5.62 seconds per epoch, in contrast to LNGD, which took only 5.08 seconds. This highlights the effectiveness of our new factorization scheme. We will explore the application of the layer-wise sampling and learning rate adjustments to other natural gradient descent methods and hope to incorporate pertinent content in the final version.
> > >
> > > Q 3. KFAC is more accurate than LNGD.
> > >
> > > A 3. Regarding the Fisher information matrix (FIM), KFAC can indeed provide a more accurate approximation compared to LNGD. However, the purpose of proposing the LNGD optimizer is to achieve an optimal balance between the training speed and accuracy of one model, and to make every effort to reduce computational cost while preserving the main information of the FIM as much as possible. Therefore, the approximation accuracy of LNGD to FIM may not be the highest, but LNGD achieves a "close-enough" approximation much faster.
> > >
> > > Q 4. Compute the Hessian matrix.
> > >
> > > A 4. In the context of LNGD, computation of the Hessian matrix is not required. When calculating the adaptive learning rate within LNGD, we approximate the Hessian matrix using the Fisher Information Matrix (FIM). This is justified by the equivalence of the FIM and the Hessian matrix under various loss functions, such as cross-entropy and mean squared loss. Accordingly, we compute the FIM and employ Eq. (12) to derive the adaptive learning rate for LNGD. However, if our proposed adaptive layer-wise learning rate is applied  to other methods without FIM, the Hessian matrix or its approximation is need to be computed.

---

> ### Comment · Reviewer_gNTY · 2024-08-13
>
> Thank you for your clarifying response, it is much appreciated. I just had a few more questions:
>
> - For the 91% and 75.9% benchmarks, can you provide references to the literature where these numbers are utilized? I'm so sorry, but I'm having trouble find it.
>
> - I'm still a bit uncomfortable with the "bundling" of many methods, and not providing similar "boosts" to the other methods in the literature. It does seem like multiple new methods are being introduced in the paper. Thus, I think it would be more scientifically enlightening to have a study where these new methods also apply to previous methods in the literature, where applicable. This is so the contribution of each part can be examined, and thus the causes of the performance increases can be made more clear.
>
> Thank you again, and I do find the research direction and the method quite interesting. And please correct me if I missed something.

---

> ### Author Response · Authors · 2024-08-14
>
> Q 1. About the reference.
>
> The related work is presented as follows：
>
> [15] Kaixin Gao, Xiaolei Liu, Zhenghai Huang, Min Wang, Zidong Wang, Dachuan Xu, and Fan Yu. A trace-restricted Kronecker-factored approximation to natural gradient. In Proceedings of the AAAI Conference on Artificial Intelligence, volume 35, pages 7519-7527, 2021.
>
> [16] Kazuki Osawa, Yohei Tsuji, Yuichiro Ueno, Akira Naruse, Chuan-Sheng Foo, and Rio Yokota. Scalable and practical natural gradient for large-scale deep learning. IEEE Transactions on Pattern Analysis and Machine Intelligence, 44(1):404-415, 2020.
>
> [17] Minghan Yang, Dong Xu, Qiwen Cui, Zaiwen Wen, and Pengxiang Xu. An efficient Fisher matrix approximation method for large-scale neural network optimization. IEEE Transactions on Pattern Analysis and Machine Intelligence, 45(5):5391-5403, 2022.
>
> [22] Thomas George, César Laurent, Xavier Bouthillier, Nicolas Ballas, and Pascal Vincent. Fast approximate natural gradient descent in a Kronecker factored eigenbasis. In Advances in Neural Information Processing Systems, pages 9550-9560, 2018.
>
> [24] Mengyun Chen, Kaixin Gao, Xiaolei Liu, Zidong Wang, Ningxi Ni, Qian Zhang, Lei Chen, Chao Ding, Zhenghai Huang, Min Wang, et al. THOR, trace-based hardware-driven layer-oriented natural gradient descent computation. In Proceedings of the AAAI Conference on Artificial Intelligence, volume 35, pages 7046-7054, 2021.
>
> Q 2. About the integration of the newly proposed strategies into existing NGD methods.
>
>
> Thank you for your response. As previously mentioned, layer-wise sampling and adaptive layer-wise learning rates can indeed be integrated with KFAC and its variants, such as EKFAC and TKFAC. However, it is important to note that the incorporation of these techniques into KFAC, EKFAC, and TKFAC results in new variants of these methodologies. Consequently, it would not be appropriate to conduct comparative experiments using these resulting new variants against LNGD. Actually, to facilitate a comprehensive understanding of the distinct contributions of various components within the LNGD framework, we have conducted a series of ablation studies, detailed in Appendix F.3 of our manuscript. These experiments aim to isolate the effects of adaptive learning rates and sampling optimization on LNGD performance.
> Furthermore, as you suggested, for a more thorough analysis of the impact of layer-wise sampling and adaptive layer-wise learning rates as independent components on the performance enhancement of existing methods such as EKFAC and TKFAC, it would be beneficial to incorporate these techniques into EKFAC and TKFAC and compare their performance against the baseline.

---

> ### Author Response · Authors · 2024-08-14
>
> Dear reviewer:
>
> We would like to express our gratitude for all the feedback provided. Additionally, we hope that the additional responses above address your concerns and contribute to an enhancement in your overall rating of this manuscript.

---

### Official Review · Reviewer_k9Mw · 2024-07-12

**Soundness:** 2
**Presentation:** 3
**Contribution:** 3
**Rating:** 7
**Confidence:** 4

**Summary:**

The authors propose a computationally feasible second-order method for training neural nets, layer-wise natural GD, which includes an Adaptive Layer-Wise Learning Rate scheme. The method eliminates the backprop pass by using a layer-local sampling approach to approximate the Fisher information matrix; thus providing a novel approach to *local learning*.  The authors provide a coherent and reasonably thourough theoretical analysis and motivation for their method, which is backed up by a *somewhat modest amount pf experimental results*.

**Strengths:**

The paper is generally well-written with only few grammatical errors/typos. The authors did a good job of presenting the math and theoretical foundation for the method (including proofs in the appendix). Clearly, a good amount of work was done. While the authors frame their work as a computationally feasible second order method, I find that it is indeed an interesting addition to the lit on local learning.

**Weaknesses:**

While the authors claim (in the abstract) to provide ``extensive experiments'', I find the work lacking wrt. empirical validation. Only three results are provided, and those are for old architectures (far from SOTA), and without error bars. The proposed method could easily have been tested on a much wider variety of (even smaller) models and benchmarks --- providing the reader with a much more complete picture of how the method behaves in priactice. This leaves a good amount of doubt wrt. the robustness of the approach.

Given the known issues with other local learning methods, I suspect that this method too might have a strong tendency to overfit. Three experiments are not sufficient to convince me otherwise.

## Speaking of local learning, the authors did not touch on the topic at all. Thus, several relevant references were missed, such as:
* Nøkland, A. (2016). Direct Feedback Alignment Provides Learning in Deep Neural Networks. ArXiv Preprint ArXiv:1609.01596. http://arxiv.org/abs/1609.01596
* Nøkland, A., & Eidnes, L. H. (2019). Training Neural Networks with Local Error Signals. Proceedings of the 36th International Conference on Machine Learning, 4839–4850. https://proceedings.mlr.press/v97/nokland19a.html
* Belilovsky, E., Eickenberg, M., & Oyallon, E. (2019). Greedy Layerwise Learning Can Scale To ImageNet. Proceedings of the 36th International Conference on Machine Learning, 583–593. https://proceedings.mlr.press/v97/belilovsky19a.html
* Belilovsky, E., Eickenberg, M., & Oyallon, E. (2020). Decoupled Greedy Learning of CNNs. Proceedings of the 37th International Conference on Machine Learning, 736–745. https://proceedings.mlr.press/v119/belilovsky20a.html
* Ren, M., Kornblith, S., Liao, R., & Hinton, G. (2022). Scaling Forward Gradient With Local Los
* Xiong, Y., Ren, M., & Urtasun, R. (2020). LoCo: Local Contrastive Representation Learning. Advances in Neural Information Processing Systems, 33, 11142–11153.ses. https://arxiv.org/abs/2210.03310v3
* Wang, Y., Ni, Z., Song, S., Yang, L., & Huang, G. (2021). Revisiting Locally Supervised Learning: an Alternative to End-to-end Training. ICLR 2021 - 9th International Conference on Learning Representations. https://arxiv.org/abs/2101.10832v1

Moreover, I really think that the authors should have included their code in the submission. Their method substantially differs from the standard SGD variants, and the code would help a lot wrt. reproducibility.

**Questions:**

* Have you considered how your method might affect the well-known implicit regularization of GD training? Specifically, the implicit bias of depth? It seems to me that skipping the backward pass might significantly affect this.
* Please explicitly define the vector, **v**, of Thm 2 (near line 255). It is only defined in the appendix.
* In Algo 1 you use both T_FIM and T_Fisherinformationmatrix. Please be consistent.

**Limitations:**

* What is your justifcation for calling your three experiments ``extensive''? Am I being too grumpy here? ;-)

---

> ### Author Rebuttal · Authors · 2024-08-07
>
> Q1: Weaknesses.
>
> A1: We would like to express our gratitude for the insightful suggestions provided. Our primary aim is to establish an optimal equilibrium between model training speed and accuracy. Building upon the existing approaches in NGD, we propose a layer-wise sampling methodology to efficiently compute the block matrix corresponding to each layer, as well as an adaptive layer-wise learning rate to enhance training efficiency. In Appendix B of our manuscript, we offer comparisons of LNGD and KFAC alongside its recent variants (EKFAC and TKFAC). To further validate the effectiveness of LNGD, we conducted additional comparative experiments on CIFAR-10 and ImageNet using recent NDG methods, consistently confirming LNGD's superior performance in achieving the same level of accuracy. Detailed experimental results are available in the response to all reviewers. Throughout the experimental process, extensive and multiple rounds of hyperparameter search were conducted, and the best results were chosen for comparison, therefore no error bars were included. As for the code, we are currently undergoing the approval process for the open-source workflow in compliance with the company's regulations, and it will be open-sourced shortly.
>
> Q2:  Have you considered how your method might affect the well-known implicit regularization of GD training? Specifically, the implicit bias of depth? It seems to me that skipping the backward pass might significantly affect this.
>
> A2: Thanks for your question. In LNGD, the key in parameter updating strategy is to calculate the updating direction ${\bf d}^k_l=({\bf F}^k_l)^{-1}\nabla_{\theta}h(\theta^k_l)$. To save computational costs, we propose the layer-wise sample approximation approach to compute ${\bf F}^k_l$ without having to perform a complete back-propagation. However, the calculation of the first-order gradient $\nabla_{\theta}h(\theta^k_l)$ still requires performing a complete back-propagation. This may not have a significant impact on the implicit bias of depth. Since we have not focused on the implicit regularization in this paper, we may not provide a complete and detailed explanation. These contents will be further considered in our following work. Thank you again for providing a valuable direction for our further study.
>
> Q3:  Please explicitly define the vector, v, of Thm 2 (near line 255). It is only defined in the appendix.
> In Algo 1 you use both T_FIM and T_Fisherinformationmatrix. Please be consistent.
>
> A3: Thanks for your questions. We will add the definition of $\bf{v}$ before Theorem 2 and correct $T_{Fisherinformationmatrix}$ as $T_{FIM}$ in the revised version.

---

> > ### Comment · Area_Chair_QGPu · 2024-08-12
> >
> > Dear Authors,
> >
> > Since Reviewer k9Mw has asked, could you please explain why the codes were not included in the submission? Also why weren't the error bars created in the plots?
> >
> > Thank you,
> > AC

---

> > > ### Author Response · Authors · 2024-08-13
> > >
> > > Regarding the open-sourcing of the code, as we are affiliated with an internet company, we initiated the process for code release upon receiving the first round of review comments. Given the multiple stages of the internal workflow, we are currently at the final stage of this process. We assure the reviewers and the area chair that the code will be made publicly available immediately upon completion of this final step.
> > >
> > > In regard to the error bars, we conducted multiple rounds of hyperparameter searches throughout the experimental process and selected the best results for final comparison; consequently, no error bars were included. If quite necessary, we can incorporate error bars in the final version.

---

> > ### Comment · Reviewer_k9Mw · 2024-08-12
> >
> > Dear Authors,
> >
> > Thanks for your careful response.
> >
> > I still *highly recommend* sharing your code. Also, please do not say "extensive experiments" in your paper; you are still far from that in my opinion ;-)
> >
> > I maintain my original rating of "Accept".

---

> > > ### Author Response · Authors · 2024-08-13
> > >
> > > Thank you for your valuable suggestion. Regarding the open-sourcing of the code, as we are affiliated with an internet company, we initiated the process for code release upon receiving the first round of review comments. Given the multiple stages of the internal workflow, we are currently at the final stage of this process. We assure the reviewers and the area chair that the code will be made publicly available immediately upon completion of this final step.
> > >
> > > In regard to ambiguous descriptions such as "extensive experiments," we will revise and refine these aspects in the final version. We sincerely apologize for any confusion caused.

---

> ### Comment · Area_Chair_QGPu · 2024-08-12
>
> Dear Reviewer k9Mw,
>
> Could you please respond with how you think about the authors' response? Please at least indicate that you have read their responses.
>
> Thank you,
> Area chair

---

### Official Review · Reviewer_HWh2 · 2024-07-13

**Soundness:** 3
**Presentation:** 2
**Contribution:** 2
**Rating:** 4
**Confidence:** 4

**Summary:**

The paper introduces a new optimization algorithm called LNGD (Layer-wise Natural Gradient Descent). This optimizer aims to enhance the training efficiency of deep neural networks by approximating the Fisher information matrix in a computationally efficient manner and introducing adaptive layer-wise learning rates.

**Strengths:**

- **Research Significance**: The paper addresses a highly valuable problem in the field of deep learning training. Improving the efficiency and effectiveness of training deep neural networks with second-order methods.
- **Practical Utility**: The proposed LNGD method shows practical promise. By reducing computational costs and introducing adaptive learning rates, the method can potentially be applied to many real-world training scenarios rather than KFAC.

**Weaknesses:**

1. **Assumption of Gaussian Distribution**: The assumption that $P_{a_l \mid a_{l-1}}$ follows a standard Gaussian distribution lacks strong motivation and empirical evidence. Even for a single layer in a random linear neural network, this assumption is not convincingly justified. Further explanation or practical case demonstrations where $P_{a_l \mid a_{l-1}}$  follows or approximates a Gaussian distribution are necessary.
2. **Convergence Analysis**: The convergence analysis results are unusual. Typically, new optimization algorithms share the relationship between convergence speed and iteration steps, assuming a certain smoothness. The approach result, similar to NTK, is unconventional and may not highlight the superiority of NGD since different optimization algorithms tend to have similar rates with a large number of hidden units.
3. **Complexity of Adaptive Learning Rate**: The method for calculating the adaptive learning rate is overly complicated. It needs to be shown whether the proposed Adaptive Layer-Wise Learning Rate has practical advantages over optimizers like Adam or LAMB. Ablation studies comparing this approach with existing LR schedules (e.g., linear warmup + cosine decay) should be conducted.
4. **Computational Overhead**: The computation of sample-wise metrics in Eq. (13) poses significant memory and computational overhead. It is necessary to verify if the final equation holds true, specifically whether the squared term's batch average is equivalent to the mean of the squared terms.
5. **Comparison with Latest Methods**: The experiments lack comparisons with some of the latest KFAC-variant methods. Additionally, current results omit several key metrics, such as peak memory usage, and selected baselines appear weak. For example, advanced training recipes for smaller models (like ResNet-50) on ImageNet typically achieve over 80% accuracy, making comparisons with sub-75% settings less meaningful.

Minor Comments
- **Layer Outputs**: Clarify the relationship between the outputs of the previous layer and the inputs of the next layer, as this affects the solution for $g_l$ in Eq. (3).
- **Notation Consistency**: The notation $F_ {LNGD}$ is not defined.
- **Citations**: The citations appear to be added post hoc, with critical parts lacking references. The related work section is also absent, suggesting a possible unfamiliarity with the latest advancements in the field, as evidenced by the absence of citations from 2023 onward.
- **Typos**: There are typographical errors in the paper, such as inconsistent matrix multiplication dimensions between lines 195-197 and missing add sign  between Lines 183-184 and
- **Algorithm Description**: The description of Algorithm 1 should not be placed in the appendix.

**Questions:**

See Weaknesses.

**Limitations:**

See Weaknesses.

---

> ### Author Rebuttal · Authors · 2024-08-07
>
> Q1:  Assumption of Gaussian distribution.
>
> A1: Thanks for your valuable suggestion. We have added Figure 1 to illustrate the validity of the Gaussian distribution assumption. Please refer to the submitted pdf file. We collect the output of two layers of the ResNet-18 network on CIFAR-10. Figure 1 (a) and (b) show the distributions of sample representation vectors' values in some dimension. Since we use the ReLU activation function, the obtained distributions are in accord with the Gaussian distribution in the positive quadrant. Figure 1 (c) and (d) show the distributions of  values of sample representation vectors' Euclidean norm, from which we can see that the two distributions can also be approximated as Gaussian distributions. We will also consider this problem further and add relevant explanations in the revised version.
>
> Q2: Convergence analysis.
>
> A2: Thank you for your insightful feedback. The convergence analysis presented in our manuscript is predominantly grounded in the framework utilized for the convergence analysis of natural gradient descent in related works. As you correctly pointed out, this result does have limitations, as different methods may exhibit similar rates when applied to a large number of hidden units. We have recognized this issue and have endeavored to provide a convergence analysis in terms of convergence rate and iteration steps within the framework of stochastic optimization with rigorous and appropriate assumptions. This work is currently ongoing and we intend to present such results in the following work in order to comprehensively illustrate the advantages of natural gradient descent in comparison with other methods.
>
> Q3: Complexity of adaptive learning rate.
>
> A3: We apologize for any ambiguity in the previous description. The adaptive learning rate is presented in Eq. (12). Furthermore, to enhance computational efficiency, we transform matrix computations into vector form as delineated in Eq. (13).  Notably, both ${\bf d}^k_l$ and ${\bf F}_l^k$  are common components in Natural Gradient Descent (NGD) approaches, ensuring that our method does not introduce additional variables but instead incorporates vector computations. In Theorem 1, we have demonstrated that the adaptive layer-wise learning rate contributes to a more rapid convergence of learnable parameters. Additionally, we conducted an Ablation Analysis in Section F.3 of the appendix, wherein confirm that the adaptive layer-wise learning rate effectively accelerates the training procedures.
>
>
> In addition, LNGD serves as a second-order optimization algorithm. In practical applications, it operates similarly to other second-order optimizers (e.g., KFAC, EKFAC and  TKFAC) and first-order optimization algorithms (e.g., Adam), which target to efficiently modify the gradient descending magnitude of training parameters and often necessitate the integration of various learning rate adjustment schedules, including the linear warmup and cosine decay strategies as you mentioned. In our manuscript, we reference other second-order optimization algorithms and incorporate an exponential updating strategy for the learning rate.
>
> Q4: Computational overhead.
>
> A4:   The aim of Eq. (13) is to expedite the computation of the adaptive layer-wise learning rate provided in Eq. (12) by converting the matrix computations to vector form. It is noteworthy that the terms ${\bf d}_l^k$ and $\mathcal{D}\theta_l$ in Eq. (13) are common variables in the NGD method, therefore we have not introduced additional parameters, but have indeed incorporated additional vector multiplication calculations to yield the adaptive layer-wise learning rate. Nonetheless, through Theorem.1 and Ablation Analysis in section F.3 in the appendix of our manuscript, the additional calculations introduced for computing the adaptive layer-wise learning rate are acceptable and can significantly speed up the convergence of training procedures.
>
>
> Regarding the validity of the final equation in Eq. (13), indeed, the strict equality does not hold. We are very grateful for your careful reading and correction. In reality, an approximate equality should be used here, where we utilize the batch average of the squared term to estimate the mean of the squared terms in computation.  We will amend this in the revised version. Thank you once again.
>
> Q5: Comparison with latest methods.
>
> A5: We acknowledge and appreciate the valuable suggestion provided. Our objective is to achieve a satisfactory balance between model training speed and accuracy. Building upon the existing NDG approaches, we propose a layer-wise sample method to efficiently compute each block matrix corresponding to each layer and introduce an adaptive layer-wise learning rate to expedite training. In Appendix B of our manuscript, we present comparisons of LNGD and KFAC along with its recent variants (EKFAC and TKFAC). To further validate the effectiveness of LNGD, we conducted additional comparative experiments, which affirm LNGD's superior performance at achieving the same level of accuracy. The detailed experimental results can be found in the response to all reviewers.  As our emphasis is not on whether LNGD can ultimately achieve the highest accuracy, but rather on the performance of various optimizations in achieving acceptable accuracy, we may have overlooked metrics such as peak memory usage. If time permits, we will evaluate the final accuracy level achievable by LNGD in the final version. Once again, we express our gratitude for the important and valuable suggestions provided for the further improvement of LNGD.
>
> Q6: Minor comments.
>
> A6: Thank you for your insightful suggestions. We will diligently proofread the manuscript and address the notations and typographical errors. Additionally, we will incorporate a comprehensive overview of recent pertinent literature published in 2023 and 2024 in the revised version. The details of Algorithm 1 will also be presented in subsection 3.3.

---

> ### Comment · Area_Chair_QGPu · 2024-08-12
>
> Dear Reviewer HWh2,
>
> Could you please respond with how you think about the authors' response? Please at least indicate that you have read their responses.
>
> Thank you, Area chair

---

> ### Author Response · Authors · 2024-08-14
>
> Dear reviewer:
>
> We sincerely appreciate your taking the time to review our response. We hope that the above clarifications address your concerns and contribute positively to your overall assessment of the paper.

---

### Author Rebuttal · Authors · 2024-08-07

We are very grateful to the four reviewers for their constructive comments and valuable suggestions on our manuscript.

The tables and figures mentioned in the reply are given in the the submitted pdf file. Please see it for details.

In our manuscript, we aim to propose an optimizer that can achieve a good balance between the training speed and accuracy of the model. To this end, based on the existing NDG approaches, we introduce LNGD. In Appendix B of our manuscript, we provide comparisons of the primary variances between LNGD, KFAC, and its recent variants (EKFAC and TKFAC). In order to further validate the effectiveness of LNGD, we conducted additional experiments on the CIFAR-10 dataset, in which three methods including EKFAC [1], TKFAC [2], and NG+ [3] are added for comparison. The detailed statistics are presented in Table 1. From the table, we observe that LNGD achieves a testing accuracy of 91\% with the fewest epochs and the shortest total time. Furthermore, LNGD exhibits the smallest computational time per epoch. Additionally, due to the efficient FIM approximation strategy adopted by NG+, it can significantly reduce the computational time and number of epochs compared to EKFAC and TKFAC. Thus, considering the time constraints, we prioritize testing the performance of NG+ on ImageNet, and the results are presented in Table 2. It is evident that LNGD consistently demonstrates a noteworthy reduction in computational time compared to NG+.

References:

[1] Thomas George, César Laurent, Xavier Bouthillier, Nicolas Ballas, and Pascal Vincent. Fast approximate natural gradient descent in a Kronecker factored eigenbasis. In Advances in Neural Information Processing Systems, pages 9550–9560, 2018.

[2] Kaixin Gao, Xiaolei Liu, Zhenghai Huang, Min Wang, Zidong Wang, Dachuan Xu, and Fan Yu. A trace-restricted Kronecker-factored approximation to natural gradient. In Proceedings of the AAAI Conference on Artificial Intelligence, volume 35, pages 7519–7527, 2021.

[3] Minghan Yang, Dong Xu, Qiwen Cui, Zaiwen Wen, and Pengxiang Xu. An efficient Fisher matrix approximation method for large-scale neural network optimization. IEEE Transactions on Pattern Analysis and Machine Intelligence, 45(5):5391–5403, 2023.

---

### Decision · Program_Chairs · 2024-09-25

**Decision:**

Accept (poster)

**Comment:**

The paper proposes a new second-order method for training neural networks using layer-wise natural gradient descent. The key design in their method is an Adaptive Layer-Wise Learning Rate scheme, where a layer-local sampling approach is used to approximate the Fisher information matrix to improve local learning. This approximation method is novel and efficient. The proposed methods have been proved effective in both theory and numerical experiments on challenging datasets.